# THE CRITIC AS AN EXPLORER: LIGHTWEIGHT AND PROVABLY EFFICIENT EXPLORATION FOR DEEP REINFORCEMENT LEARNING

## ABSTRACT

Exploration remains a critical challenge in reinforcement learning (RL), with many existing methods either lacking theoretical guarantees or being computationally impractical for real-world applications. We introduce *Litee*, a lightweight algorithm that repurposes the value network in standard deep RL algorithms to effectively drive exploration without introducing additional parameters. *Litee* utilizes linear multi-armed bandit (MAB) techniques, enabling efficient exploration with provable sub-linear regret bounds while preserving the core structure of existing RL algorithms. *Litee* is simple to implement, requiring only around 10 lines of code. It also substantially reduces computational overhead compared to previous theoretically grounded methods, lowering the complexity from $O(n^3)$ to $O(d^3)$, where $n$ is the number of network parameters and $d$ is the size of the embedding in the value network. Furthermore, we propose *Litee+*, an extension that adds a small auxiliary network to better handle sparse reward environments, with only a minor increase in parameter count (less than 1%) and additional 10 lines of code. Experiments on the MiniHack suite and MuJoCo demonstrate that *Litee* and *Litee+* empirically outperform state-of-the-art baselines, effectively bridging the gap between theoretical rigor and practical efficiency in RL exploration.

## 1 INTRODUCTION

Exploration remains a fundamental challenge in reinforcement learning (RL), particularly in environments with sparse rewards or complex dynamics. Although algorithms such as DQN [26], PPO [34], SAC [13], DDPG [24], TD3 [12], and IMPALA [10] have demonstrated impressive performance on tasks like Atari games [25; 26], StarCraft [37], and Go [35], they often depend on rudimentary exploration strategies. Common approaches, such as $\epsilon$-greedy policies or injecting noise into actions, are typically inefficient and can struggle in scenarios with delayed or sparse rewards.

Various exploration methods have been proposed to improve performance and address the challenge of reward sparsity. For decades, exploration strategies with proven optimality in tabular settings have been available [20]. More recently, methods with provable regret bounds have been developed for scenarios involving function approximation, including linear functions [27; 28; 18; 19; 1], kernels [40], and neural networks [40]. However, while linear and kernel-based approaches make strong assumptions about the structure of RL functions, provable methods based on neural networks often suffer from prohibitive computational costs—specifically $O(n^3)$ complexity, where $n$ is the number of parameters in the RL network—making these methods impractical for real-world applications.

A more practical approach to exploration relies on heuristics, leading to the development of several empirically successful methods, such as Pseudocount [5], ICM [29], RND [6], RIDE [30], NovelD [42], AGAC [11], and E3B [14; 15]. These methods typically use internally generated bonuses to incentivize agents to explore novel states based on specific metrics. For instance, RND [6] utilizes the prediction error of a randomly initialized target network as the exploration bonus, while RIDE [30] combines the errors from forward and inverse dynamics models. However, these methods lack theoretical guarantees and are primarily driven by intuitive heuristics. Furthermore, they often require the training of additional networks beyond the standard value or policy networks in RL algorithms, which makes them computationally expensive.

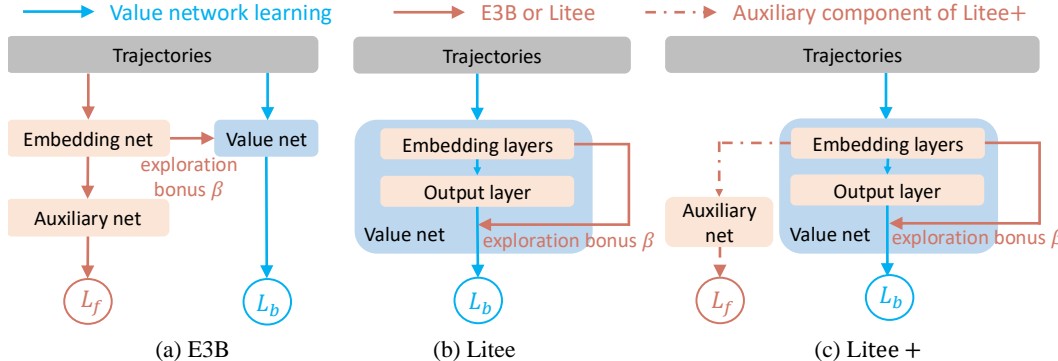

Figure 1: Comparison between a representative exploration approach (a) E3B [14] , (b) *Litee*, and (c) *Litee+*. E3B requires additional networks to generate exploration bonuses, while *Litee* repurposes the value network's state embeddings, resulting in reduced computational overhead and no additional parameters. *Litee+* extends *Litee* by incorporating a small auxiliary network to enhance performance in sparse reward environments, with only a minor increase in parameters.

In this work, we aim to combine the strengths of both theoretically grounded and empirically effective exploration methods. Provably efficient exploration strategies that leverage function approximation [18; 19; 40; 27; 28; 32] are fundamentally rooted in the theory of contextual Multi-Armed Bandits (MAB) [22; 9; 2; 38; 43; 44]. Building on this foundation, we hypothesize that advanced techniques from *neural MAB* can be effectively adapted for exploration in *deep RL*. Empirical results indicate that decoupling deep representation learning from exploration strategies, such as Upper Confidence Bound (UCB) or Thompson Sampling in linear MAB [41; 31; 39], shows promise for achieving efficient exploration in neural MAB.

Motivated by these insights, we propose ***Litee***: a **Lite e**xploration algorithm for deep RL. Unlike existing methods [5; 6; 29; 29; 30; 14; 15], which require training additional embedding networks for state representation, *Litee* directly utilizes the state embeddings of the existing value network in the RL algorithm, applying linear MAB techniques for exploration. As a result, *Litee* introduces no new parameters beyond those already present in the original algorithm, demonstrating that RL algorithms inherently possess strong exploration capabilities when their learned networks are effectively leveraged. Moreover, *Litee* is simple to implement—requiring only around 10 lines of code. For more complex tasks, where learning from sparse rewards is especially challenging, *Litee* can be enhanced by incorporating a small auxiliary network to accelerate the learning process. This extended version, *Litee+*, results in only a minimal increase in parameter count (less than 1%) and implementation effort (approximately 10 additional lines of code).

We evaluated *Litee+* and *Litee* on the MiniHack and MuJoCo benchmarks to assess their effectiveness in both sparse and dense reward environments. *Litee* either outperforms or at least matches the performance of state-of-the-art baseline methods such as PPO [34], SAC [13] and TD3 [12], which are not specifically designed for exploration. In contrast, *Litee+* consistently outperforms E3B [14], the state-of-the-art exploration method for MiniHack, across all evaluated tasks, demonstrating superior reliability and effectiveness in diverse reinforcement learning settings.

In summary, we make three key contributions in this paper. First, we propose ***Litee***, a lightweight exploration algorithm that integrates seamlessly with existing RL algorithms without introducing additional parameters, and extend it to ***Litee+*** for improved performance in sparse-reward environments. Second, we provide theoretical guarantees, showing that any RL algorithm enhanced with ***Litee*** achieves a sub-linear regret bound over episodes. Finally, we validate the effectiveness of *Litee* and *Litee+* through experiments on the MiniHack and MuJoCo benchmarks, demonstrating their superior performance in both sparse and dense reward settings.

## 2 RELATED WORK

**Multi-Armed Bandits**. MAB algorithms address the exploitation-exploration dilemma by making decisions and receiving rewards over time under uncertainty. LinUCB [22] assumes linearity in re-

Table 1: Comparison of exploration methods on MiniHack tasks. **Networks**: additional networks required beyond those in IMPALA, which contains $25,466,652$ parameters; **parameters**: the number of additional parameters introduced by the exploration module; $\uparrow$ means the percentage of parameter increase. Networks in **bold** represent those with significant parameters, while those in gray indicate substantially fewer parameters. *Litee+* refers to *Litee* with the small auxiliary network added.

| Algorithm | Networks | parameters | $\uparrow$ (%) |
|---|---|---|---|
| ICM | **Embedding net** + Forward dynamics net + Inverse dynamics net | $16,074,512 + 2,110,464 + 527,371$ | 73% |
| RND | **Embedding net** | $16,074,512$ | 63% |
| RIDE | **Embedding net** + Forward dynamics net + Inverse dynamics net | $16,074,512 + 2,110,464 + 527,371$ | 73% |
| NovelD | **Embedding net** | $16,074,512$ | 63% |
| E3B | **Embedding net** + Inverse dynamics net | $16,074,512 + 527,371$ | 65% |
| *Litee* | - | - | 0% |
| *Litee+* | Inverse dynamics net | $199,819$ | 0.8% |

wards concerning arm contexts and guarantees a sub-linear regret bound [9]. To relax the linearity assumption, KernelUCB [36; 8] and NegUCB [23] map contexts to high-dimensional spaces and apply LinUCB in these transformed settings. Neural-UCB [44] and Neural-TS [43] utilize neural networks to model the relationship between contexts and rewards, though their computation time of $O(n^3)$, where $n$ is the number of network parameters, limits their scalability in real-world tasks. Neural-LinTS [31] and Neural-LinUCB [39] effectively decouple representation learning from exploration, enhancing the practicality of network-based bandit algorithms.

**Exploration in RL**. Common exploration strategies in RL, such as $\epsilon$-greedy [26] and stochastic noise [24; 34], often lack sample efficiency and struggle with sparse rewards. While provably sample-efficient algorithms [20; 27; 28; 18; 19; 1; 7] based on MAB theory exist, they face empirical limitations or are primarily theoretical, lacking practical applicability in deep RL [4]. Many successful empirical methods [5; 29; 6; 30; 42; 11; 14; 15] rely on exploration bonuses that incentivize agents to visit novel states, but these approaches often lack theoretical grounding and require training significantly more parameters. In contrast, *Litee* utilizes MAB methods for exploration, assisted by embedding layers within the RL value network, providing empirical benefits with minimal additional parameters. Figure 1 illustrates the differences between E3B and *Litee*, while Table 1 summarizes the additional networks and parameters of various exploration methods.

# 3 METHODOLOGY

Unless otherwise specified, bold uppercase symbols denote matrices, while bold lowercase symbols represent vectors. $\boldsymbol{I}$ refers to an identity matrix, and $\boldsymbol{0}$ represents a zero vector. Frobenius norm and $l_2$ norm are both denoted by $\|\cdot\|_2$. Mahalanobis norm of a vector $\boldsymbol{x}$ based on matrix $\boldsymbol{A}$ is given by $\|\boldsymbol{x}\|_{\boldsymbol{A}} = \sqrt{\boldsymbol{x}^\top \boldsymbol{A} \boldsymbol{x}}$. For an integer $K > 0$, the set of integers $\{1, 2, ..., K\}$ is represented by $[K]$.

## 3.1 PRELIMINARY

An episodic Markov Decision Process (MDP) is formally defined as a tuple $(\mathcal{S}, \mathcal{A}, H, \mathbb{P}, r)$, where $\mathcal{S}$ denotes the state space and $\mathcal{A}$ is the action space. Integer $H > 0$ indicates the duration of each episode. Functions $\mathbb{P} : \mathcal{S} \times \mathcal{A} \times \mathcal{S} \to [0,1]$ and $r : \mathcal{S} \times \mathcal{A} \to [0,1]$ are the Markov transition and reward functions, respectively. During an episode, the agent follows a policy $\pi : \mathcal{S} \times \mathcal{A} \to [0,1]$. At each time step $h \in [H]$ in the episode, the agent observes the current state $s_h \in \mathcal{S}$ and selects an action $a_h \sim \pi(\cdot|s_h)$ to execute, then the environment transits to the next state $s_{h+1} \sim \mathbb{P}(\cdot|s_h, a_h)$, yielding an immediate reward $r_h = r(s_h, a_h)$.

Various algorithms have been developed to learn the optimal policy $\pi^*$ for the agent to select and execute actions at each time step $h$ in the episode, thus ultimately maximizing the long-term return $\sum_{h=1}^{H} \gamma^{h-1} r_h$, where $0 < \gamma < 1$ is the discount parameter. Notable algorithms include DQN [26], PPO [34], SAC [13], IMPALA [10], *etc*. A common component of these algorithms is the use of a

network to approximate the action-value function[1] $Q$ under a specific policy as Equation 1, where $\phi(\cdot, \cdot | \boldsymbol{W})$ is the embedding layers, $\boldsymbol{\theta}$ and $\boldsymbol{W}$ are trainable parameters. At step $h$ in the episode, the action-value $Q(s_h, a_h)$ approximates the long-term return $\sum_{t=h}^{H} \gamma^{t-h} r_t$ after executing action $a_h$ at state $s_h$ and following the specific policy thereafter:

$$Q(s, a) = \boldsymbol{\theta}^{\mathsf{T}} \phi(s, a | \boldsymbol{W}). \tag{1}$$

The Bellman equation [26] is employed to update the action-value function. Using the most recent action-value function, the policy can be updated in various ways, depending on the specific algorithm. Since *Litee* focuses on leveraging Equation 1 for efficient exploration while preserving the core techniques of existing algorithms, we introduce *Litee* within the context of DQN for simplicity; however, it can be easily adapted to other algorithms.

### 3.2 *Litee*: EXPLORATION WITH VALUE NETWORK UNDER UNCERTAINTY

For the state-action pair $(s_h, a_h)$ at time step $h$, the approximated action-value $Q(s_h, a_h)$ is subject to an uncertainty term $\beta(s_h, a_h)$. This uncertainty arises from the novelty or limited experience with the particular state-action pair. Similar to MAB problems, it is essential to account for this uncertainty when utilizing the latest approximated action-value function. Incorporating the uncertainty term encourages exploration, ultimately improving long-term performance. Thus, the action-value function adjusted for uncertainty is given by Equation 2, where $\alpha \geq 0$ is the exploration coefficient:

$$Q(s, a) = \boldsymbol{\theta}^{\mathsf{T}} \phi(s, a | \boldsymbol{W}) + \alpha \beta(\cdot, \cdot). \tag{2}$$

However, defining $\beta(\cdot, \cdot)$ remains a significant challenge. Traditional MAB methods often attempt to address this by either assuming a linear action-value function or relying on algorithms that require $O(n^3)$ computation time in terms of the number of parameters $n$ in the action-value network. Both of these approaches have inherent drawbacks. Linearity may fail to capture the complexity of real-world tasks. On the other hand, algorithms with cubic computation time become impractical.

To overcome these limitations, we draw inspiration from Neural-LinUCB [39] and Neural-LinTS [31], which effectively decouple representation learning from exploration. Building on this idea, *Litee* adopts a similar approach, decomposing the action-value function into two distinct components. This decomposition follows the standard value network structure (Equation 1), while providing a flexible and computationally efficient framework for balancing exploration and exploitation:

- Network $\phi(s, a | \boldsymbol{W})$ extracts the embeddings of state-action pair $(s; a)$;
- $Q(s, a) = \boldsymbol{\theta}^{\mathsf{T}} \phi(s, a | \boldsymbol{W})$ is linear in the embedding of $(s, a)$ with parameter $\boldsymbol{\theta}$.

Consequently, MAB theory with the linearity assumption can be applied to the embedding $\phi(s, a)$ for $\forall s \in \mathcal{S}$ and $\forall a \in \mathcal{A}$. Simultaneously, the action-value function retains its representational capacity through the neural network $\phi(s, a)$, ensuring promising empirical performance.

Algorithm 1 details DQN with *Litee*[2]. In this algorithm, all lines except those highlighted in blue follow the standard DQN framework, while the blue lines specifically represent the adjustment of the action-value function to account for uncertainty. For conciseness, we denote the result of $\phi(s_h^m, a_h^m)$ as the vector $\phi_h^m$, which is assumed to be $d$-dimensional, *i.e.*, $\phi_h^m \in \mathbb{R}^d$. Algorithm 1 initializes the variance matrix as $\boldsymbol{A} = \lambda \boldsymbol{I}$ where $\lambda > 0$ is the ridge parameter. Based on the latest variance matrix, we introduce two methods to define the uncertainty term: UCB- and Thompson Sampling-based uncertainty term, each corresponding to a different exploration strategy.

**Uncertainty term based on UCB**. Upper Confidence Bound (UCB) is a widely used optimistic exploration strategy, where the agent assumes the best-case scenario in the face of uncertainty. In this approach, the uncertainty term is proportional to the estimated variance and serves as a measure

---

[1]In some algorithms, the state- instead of the action-value functions are learned. However, this does not affect the implementation and conclusion of our method, as will be seen in Section 3.2.

[2]It is a concise version for easier comprehension. In Appendix B, we present the complete version in Algorithm 3.

---

**Algorithm 1** Deep Q-Network (DQN) with *Litee*. The lines highlighted in blue represent modifications that introduce *Litee*'s exploration enhancements, incorporating uncertainty estimation and variance updates to improve exploration efficiency.

---

1: **Input:** Ridge parameter $\lambda > 0$, the exploration parameter $\alpha \geq 0$, episode length $H$, episode number $K$
2: **Initialize:** Covariance matrix $\boldsymbol{A} = \lambda \boldsymbol{I}$, parameters $\boldsymbol{\theta} \sim \frac{1}{d} N(\boldsymbol{0}, \boldsymbol{I})$, networks $\phi(\cdot, \cdot | \boldsymbol{W})$ [39], the action-value network $Q = \boldsymbol{\theta}^{\mathsf{T}} \phi(s, a)$, and the target value-networks $\bar{Q}(s, a) = Q(s, a)$

3: **for** episode $m = 1$ **to** $M$ **do**
4:     Sample the initial state of the episode $s_1^m$
5:     **for** step $h = 1, 2, ..., H$ **do**
6:         Conduct action $a_h^m = \arg\max_a Q(s_h^m, a)$ and get the next state $s_{h+1}^m$ and reward $r_h^m$
7:         Update the parameters of the action-value function $\boldsymbol{\theta}$ and $\boldsymbol{W}$ by Bellman equation [26]

8:         Approximate the uncertainty term $\beta(\cdot, \cdot)$ by Equation 3 or Equation 4

9:         Approximate the action-value in the face of uncertainty $Q(s, a)$ by Equation 2

10:       Update the variance matrix $\boldsymbol{A}$ by Equation 5

11:     **end for**
12:     Update the target network $\bar{Q}(\cdot, \cdot) = Q(\cdot, \cdot), h \in [H]$
13: **end for**

---

of uncertainty in the action-value function approximation. The higher the uncertainty, the more likely the agent is to explore. As uncertainty decreases, the agent gradually shifts towards exploiting the known information for decision-making. This method defines the uncertainty term as:

$$\beta(s, a) = \sqrt{\phi(s, a)^{\mathsf{T}} \boldsymbol{A}^{-1} \phi(s, a)}. \tag{3}$$

**Uncertainty term based on Thompson Sampling**. Instead of relying on a fixed optimistic uncertainty, this approach samples from a posterior distribution over the possible value functions. By sampling from this distribution, the agent naturally balances exploration and exploitation based on the likelihood of each action being optimal. This method defines the uncertainty term as:

$$\begin{aligned} \boldsymbol{\Delta\theta} &\sim N(0, \boldsymbol{A}^{-1}), \\ \beta(s, a) &= (\boldsymbol{\Delta\theta})^{\mathsf{T}} \phi(s, a). \end{aligned} \tag{4}$$

At each time step $h$ in episode $m$, after calculating the uncertainty and approximating the actionn-value function with uncertainty, we update the variance matrix before proceeding to the next step:

$$\boldsymbol{A} = \boldsymbol{A} + \phi_h^m (\phi_h^m)^{\mathsf{T}}. \tag{5}$$

Algorithm 1 is straightforward and easy to implement, while offering several advantages over existing approaches. E3B [14] introduces a bonus term similar to that in Equation 3; however, it relies on additional networks to approximate the embedding, which is heuristic and lacks theoretical guarantees. Other approaches also incorporate MAB methods, but they typically treat the action-value function as either a linear or kernel function [19; 40], which limits their applicability to real-world tasks. Furthermore, some methods [40] require $o(n^3)$ computation time where $n$ is the number of the action-value network's parameters, making them impractical to implement. Additionally, certain approaches only provide proofs related to the MAB method while neglecting the theoretical analysis of the deep RL algorithm [4]. In contrast to these methods, Algorithm 1 does not require learning any additional parameters beyond those already present in the RL algorithms. Computationa time associated with the uncertainty term is $o(d^3)$ where $d \ll n$ represents the embedding dimension. Furthermore, it offers theoretical guarantee, which will be elaborated upon in Section 4.

**Adapting to General RL Algorithms**. To apply Algorithm 1 to general RL algorithms, we incorporate the UCB- or TS-based uncertainty into the action-value function by reshaping the immediate rewards. Additionally, depending on the algorithm employed, we may sometimes learn the state-instead of the action-value network. As a result, the value network can only derive state embeddings rather than state-action pair embeddings. Even when learning the action-value network, it may still output only state embeddings if it is designed to take states as input and produce action-values for

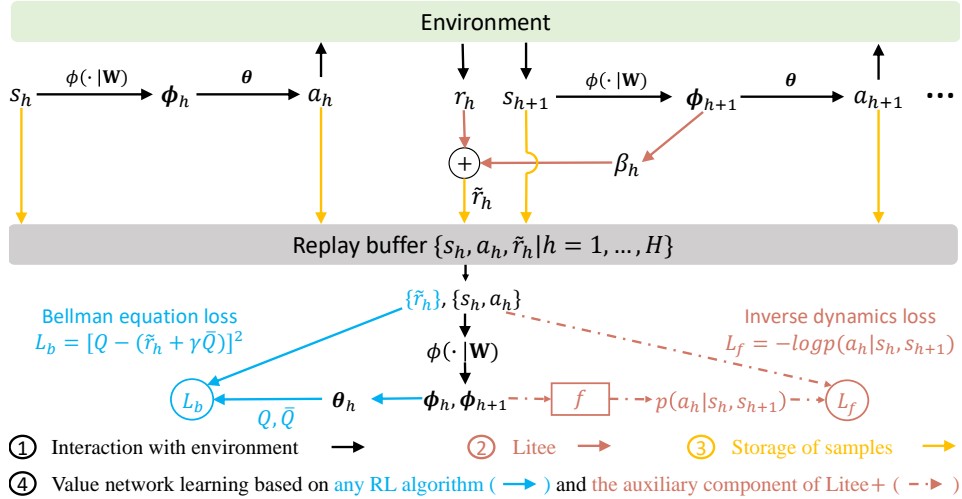

Figure 2: *Litee* framework. $L_b$ represents the Bellman loss used to update the action-value function, while $L_f$ refers to the loss of the auxiliary network, which will be detailed in Equation 6.

each action. In such cases, the embedding of the next state is utilized to replace the embedding of the current state-action pair. For notational simplicity, we continue to refer to the state embedding network as $\phi(\cdot)$ and the output $\phi(s_h^m)$ as $\phi_h^m$, assuming no ambiguity arises. As a result, the practical algorithm incorporating *Litee* is presented in Figure 2 and Algorithm 2. It can seamlessly adapt to any RL algorithm, with the only additional step being *reward shaping*.

### 3.3 *Litee+*: ENHANCING *Litee* WITH MINIMAL OVERHEAD

For tasks where learning value networks from sparse rewards is challenging, a small network can be incorporated to accelerate learning, introducing only a minimal number of additional parameters. Specifically, we utilize the Inverse Dynamics Network (IDN) [29; 30; 14] to enhance the learning of the embedding layers contained in the action-value network. This is achieved by a compact network $f$ that infers the distribution $p(a)$ over actions given consecutive states $s_h$ and $s_{h+1}$, which is trained by maximum likelihood estimation:

$$L_f = -\log p(a_h|s_h, s_{h+1}). \tag{6}$$

To introduce this enhancement with minimal additional parameters, we utilize the state embeddings $\phi(s_h)$ and $\phi(s_{h+1})$ from the value network. These embeddings are first transformed by a linear layer $u$ parameterized by $\boldsymbol{W}_u$, followed by a small network $v$, which takes the transformed consecutive embeddings to infer the corresponding action:

$$p(a_h|s_h, s_{h+1}) = f(\phi(s_h), \phi(s_{h+1})) = v(\boldsymbol{W}_u\phi(s_h), \boldsymbol{W}_u\phi(s_{h+1})). \tag{7}$$

In our design, the module $f$ is purposefully kept lightweight by significantly reducing the number of parameters compared to the value network, ensuring minimal computational overhead. To further enhance efficiency, we update the embedding in Line 7 of Algorithm 2 as $\phi_h^m = \boldsymbol{W}_u\phi(s_{h+1}^m)$.

This design brings several advantages. First, the introduction of $u$ effectively decouples the policy from the Inverse Dynamics Network (IDN), reducing interdependencies that could hinder learning and thereby improving empirical performance. Second, since $u$ is a simple linear transformation of $\phi(s_{h+1})$, it also retains the theoretical guarantees of UCB- and Thompson Sampling-based exploration strategies, maintaining the rigor and stability of the exploration process. Third, transforming $\phi_h^m$ into a lower-dimensional embedding with $\tilde{d} < d$ not only reduces the number of additional parameters but also brings down the computational complexity of $\beta_h^m$ to $o(\tilde{d}^3)$, making the method computationally efficient and scalable for practical applications.

Notably, IDN is also applicable when the embedding network is designed for state-action pairs, *i.e.*, $\phi(s, a)$. In this case, a constant default value is used for the action, while the actual states are input, with the resulting outputs treated as the state embeddings.

---

**Algorithm 2** *Litee* for general deep RL. Either UCB-based or Thompson Sampling-based uncertainty can be used depending on the desired exploration strategy.

---

1: **Input:** Ridge parameter $\lambda > 0$, exploration parameter $\alpha \geq 0$, episode length $H$, episode number $K$
2: **Initialize:** Covariance matrix $\boldsymbol{A} = \lambda \boldsymbol{I}$, initial policy $\pi(\cdot)$, state- or action-value function $V(\cdot)$ or $Q(\cdot, \cdot)$
3: **for** episode $m = 1$ **to** $M$ **do**
4:     Receive the initial state $s_1^m$ from the environment
5:     **for** step $h = 1, 2, ..., H$ **do**
6:         Conduct action $a_h^m \sim \pi(s_h^m)$ and observe the next state $s_{h+1}^m$ and receive reward $r_h^m$
7:         Get embedding of the next state $\boldsymbol{\phi}_h^m = \phi(s_{h+1}^m)$
8:         Calculate action-value variance $b_h^m = (\boldsymbol{\phi}_h^m)^{\mathsf{T}} \boldsymbol{A}^{-1} \boldsymbol{\phi}_h^m$
9:         Generate UCB-based action-value uncertainty $\beta_h^m = \sqrt{b_h^m}$
10:       Generate Thompson Sampling-based action-value uncertainty $\beta_h^m \sim N(0, b_h^m)$
11:       Reshape the reward $r_h^m = r_h^m + \alpha \beta_h^m$
12:       Update the covariance matrix $\boldsymbol{A} = \boldsymbol{A} + \boldsymbol{\phi}_h^m (\boldsymbol{\phi}_h^m)^{\mathsf{T}}$
13:     **end for**
14:     Adopt any RL algorithm to update the value function $V(\cdot)$ or $Q(\cdot, \cdot)$ and the policy $\pi(\cdot)$
15: **end for**

---

# 4 THEORETICAL ANALYSIS

In this section[3], we introduce additional notation before delving into the detailed theory. Under the true optimal policy $\pi^*$, assume the corresponding action-value function $Q^*$ is structured as in Equation 1 and parameterized by $\boldsymbol{\theta}^*$ and $\boldsymbol{W}^*$. In Algorithm 1, the policy executed in episode $m \in [M]$ is denoted by $\pi_m$, with its action-value function represented as $Q^{\pi_m}$. Cumulative regret of Algorithm 1 is as definition 4.1.

**Definition 4.1.** *Cumulative Regret. After $M$ episodes of interactions with the environment, the cumulative regret of Algorithm 1 is defined as Equation 8, where $u_1^m$ is the optimal action at state $s_1^m$ generated by policy $\pi^*$ while $a_1^m$ is that selected by the executed policy $\pi_m$.*

$$\text{Regret}_M = \sum_{m=1}^{M} Q^*(s_1^m, u_1^m) - Q^{\pi_m}(s_1^m, a_1^m). \tag{8}$$

Cumulative regret quantifies the gap between the optimal return and the actual return accumulated over $M$ episodes of interaction with the environment. By establishing a sub-linear upper bound on Equation 8 with respect to the number of episodes $M$, we can demonstrate the sample efficiency of *Litee*. *Litee* draws inspiration from Neural-LinUCB [39] and Neural-LinTS [31], corresponding to the UCB- and Thompson Sampling-based action-value functions, respectively. The theoretical analysis of *Litee* builds on the conclusions from these methods. While Neural-LinUCB is supported by theoretical analysis, Neural-LinTS has only been validated empirically. In this paper, we present the regret bound for Neural-LinTS in Section D.2, leading us to the regret bound for Algorithm 1, as stated in Equation 4.2. The proof is deferred to Appendix C.

**Theorem 4.2.** *Suppose the standard initializations and assumptions from the literature [40; 39] hold. Furthermore, without loss of generality, assume that $\|\boldsymbol{\theta}^*\|_2 \leq 1$ and $\|(s_h; a_h)\|_2 \leq 1$. For any $\sigma \in (0, 1)$, let:*

$$\alpha = \sqrt{2(d \cdot \log(1 + \frac{M \cdot \log |\mathcal{A}|}{\lambda}) - \log \sigma)} + \sqrt{\lambda}$$

$$\eta \leq C_1 (\iota \cdot d^2 M^{\frac{11}{2}} L^6 \cdot \log \frac{M |\mathcal{A}|}{\sigma})^{-1}, \tag{9}$$

*and the number of parameters in each of the $L$ layers of $\phi(\cdot, \cdot)$ is at least $\iota = \text{poly}(L, d, \frac{1}{\sigma}, \log \frac{M|\mathcal{A}|}{\sigma})$, where $|\mathcal{A}|$ means the action space size and $\text{poly}(\cdot)$ means a polynomial function depending on the incorporated variables, then with probability at least $1 - \sigma$, it holds that:*

---

[3]Conclusions in this section are to Algorithm 3, the complete version of Algorithm 1.

$$\text{Regret}_M \leq \underbrace{C_2 \alpha H \sqrt{Md \cdot \log(1 + \frac{M}{\lambda d})} + H \sqrt{16MH \log \frac{2}{\sigma}} + H \sqrt{2MH \log \frac{2}{\sigma}}}_{\widetilde{O}(\sqrt{M})} \tag{10}$$

$$+ \frac{C_3 \cdot HL^3 d^{\frac{5}{2}} M \sqrt{\log(\iota + \frac{1}{\sigma} + \frac{M|\mathcal{A}|}{\sigma})} \|\boldsymbol{q} - \tilde{\boldsymbol{q}}\|_{\boldsymbol{H}^{-1}}}{\iota^{\frac{1}{6}}},$$

*where $C_1, C_2, C_3$ are constants independent of problem parameters; $\boldsymbol{q} = (q_1^1; q_2^1; ...; q_1^M; ...; q_H^M)$ and $\tilde{\boldsymbol{q}} = (Q_1^1(s_1^1, a_1^1); Q_1^1(s_2^1, a_2^1); ...; Q_1^M(s_1^M, a_1^M); ...; Q_H^M(s_H^M, a_H^M))$ are respectively the target and the estimated value vectors; $\boldsymbol{H}$ is the neural tangent kernel, as defined in [39].*

Specifically, in theorem 4.2, we assume $\|\boldsymbol{\theta}^*\|_2 \leq 1$ and $\|(s_h; a_h)\|_2 \leq 1$ to make the bound scale-free. Otherwise, the bound would increase by a scale factor. Neural tangent kernel $\boldsymbol{H}$ is defined in accordance with a recent line of research [17; 3] and is essential for the analysis of overparameterized neural networks. Other standard assumptions and initialization are explained in Section D.1. From Equation 10, we can conclude that the upper bound of the cumulative regret grows sub-linearly with the number of episodes $M$, *i.e.*, $\widetilde{O}(\sqrt{M})$ where $\widetilde{O}(\cdot)$ hide constant and logarithmic dependence of $M$, indicating that the executed policy improves over time. Notably, the last term in Equation 10 arises from the error due to network estimation. Here, $M$ can be traded off against $\iota$ and the estimation error $\|\boldsymbol{q} - \tilde{\boldsymbol{q}}\|_{\boldsymbol{H}^{-1}}$, making it often neglected in the literature.

## 5 EXPERIMENT

In this section, we evaluate *Litee+* and *Litee* across tasks from both MiniHack and MuJoCo, which feature sparse and dense rewards, respectively. For the MiniHack tasks, we select IMPALA as the base RL algorithm due to its status as a state-of-the-art method and its frequent use in exploration problem baselines. Given the sparse reward nature of MiniHack tasks, we choose *Litee+* and compare IMPALA with *Litee+* against six baselines: IMPALA [10], ICM [29], RND [6], RIDE [30], NovelD [42], and E3B [14]. Notably, all except IMPALA are specifically designed for sparse reward settings and also use IMPALA as their base RL algorithm. For the MuJoCo tasks, which involve dense rewards, we evaluate three state-of-the-art RL algorithms: SAC [13], PPO [34], and TD3 [12], with and without *Litee*.

**Reproducibility**. The experiments presented in this paper are based on publicly available codebases from E3B [4] [14] and CleanRL [5] [16]. To ensure reproducibility, we provide the core code and detailed hyperparameters for *Litee* and *Litee+* in Appendix E and Appendix A, respectively. In fact, the experiments can be easily replicated with minimal modifications to the provided code.

### 5.1 SPARSE REWARD TASKS

MiniHack [33] is built on the NetHack Learning Environment [21], a challenging video game where an agent navigates procedurally generated dungeons to retrieve a magical amulet. MiniHack tasks present a diverse set of challenges, such as locating and utilizing magical objects, traversing hazardous environments like lava, and battling monsters. These tasks are characterized by sparse rewards, and the state provides a wealth of information, including images, text, and more, though only a subset is relevant to the specific task at hand.

As shown in Table 1, *Litee+* adds approximately $0.8\%$ more parameters compared to IMPALA, which does not include a dedicated exploration module. In contrast, other baselines with specifically designed exploration modules, such as RIDE and E3B, introduce $60\% - 80\%$ additional parameters over IMPALA. This highlights the lightweight nature of *Litee*.

We present the experimental results for E3B, IMPALA, and *Litee+* to conserve computational resources. IMPALA serves as the baseline without a specifically designed exploration module, while

---

[4]https://github.com/facebookresearch/e3b

[5]https://github.com/vwxyzjn/cleanrl

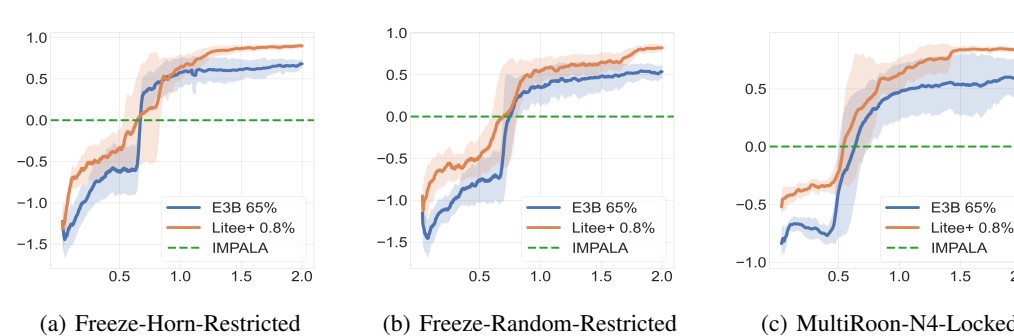

(a) Freeze-Horn-Restricted     (b) Freeze-Random-Restricted     (c) MultiRoon-N4-Locked

Figure 3: Experiment results on MiniHack over seeds $1-3$. The vertical axis represents the average return, while the horizontal axis denotes the number of frames, in multiples of $1e7$. For IMPALA, we only display its performance upper bounds, as it fails to achieve positive average scores. The legend includes the percentage of additional parameters introduced by each algorithm compared to the original network (65% increase for E3B [14] and 0.8% increase for *Litee+*).

E3B is recognized as the state-of-the-art method among exploration problem baselines on MiniHack. Results for additional baselines, including ICM, RND, RIDE, and NovelD, can be found in the E3B paper [14] and can be reproduced using the provided code. Based on previously reported findings as well as our own reproductions, these baselines typically struggle to achieve positive average scores without significant human engineering, which is one reason they are not discussed in further detail.

The experimental results presented in Figure 3(a), Figure 3(b), and Figure 3(c) correspond to three MiniHack tasks, where *Litee+* employs Thompson Sampling-based exploration. It is clear that *Litee+* consistently outperforms E3B across these various MiniHack tasks. While *Litee+* may converge slightly more slowly than E3B at times, this is expected, as *Litee+* tends to explore the environment more thoroughly before heavily exploiting its accumulated experiences. However, once convergence is achieved, *Litee+* demonstrates significantly superior performance compared to E3B. Given that E3B relies on bonus-based reward reshaping, it can be challenging to ensure that maximizing cumulative return directly aligns with maximizing the reshaped return. In contrast, *Litee+* benefits from strong theoretical guarantees regarding cumulative regret, which helps account for its robust empirical performance.

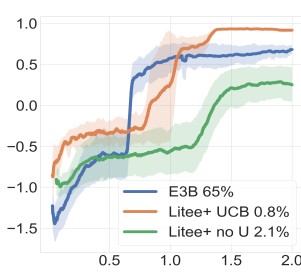

(a) Freeze-Horn-Restricted

We also implemented *Litee+* with UCB-based exploration. A comparison of the results from *Litee+* using Thompson Sampling- and UCB-based exploration, shown in Figure 3 and Figure 4, respectively, reveals that both methods yield comparable outcomes. Additionally, we conducted an ablation study on $U$, which is designed to prevent severe coupling between the policy and the IDN. As illustrated in Figure 4(a) and Figure 4(b), $U$ is crucial for enhancing the empirical performance of *Litee+*. Without $U$, *Litee+* occasionally outperforms E3B, though there are instances where it does not. Furthermore, without $U$, *Litee+* introduces a larger number of additional parameters, specifically $2.1\%$. For additional experimental results on other MiniHack tasks, please refer to Appendix E.

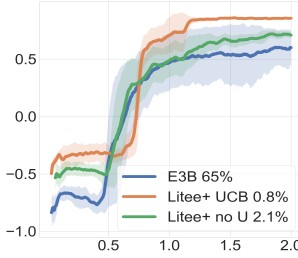

(b) MultiRoon-N4-Locked

Figure 4: Ablation study.

## 5.2 DENSE REWARD TASKS

For dense reward tasks, we utilize the MuJoCo testbed, a widely used physics-based simulation environment for benchmarking RL algorithms. MuJoCo provides a suite of continuous control tasks where agents must learn to perform various actions, such as locomotion, manipulation, and balancing, within simulated robotic environments.

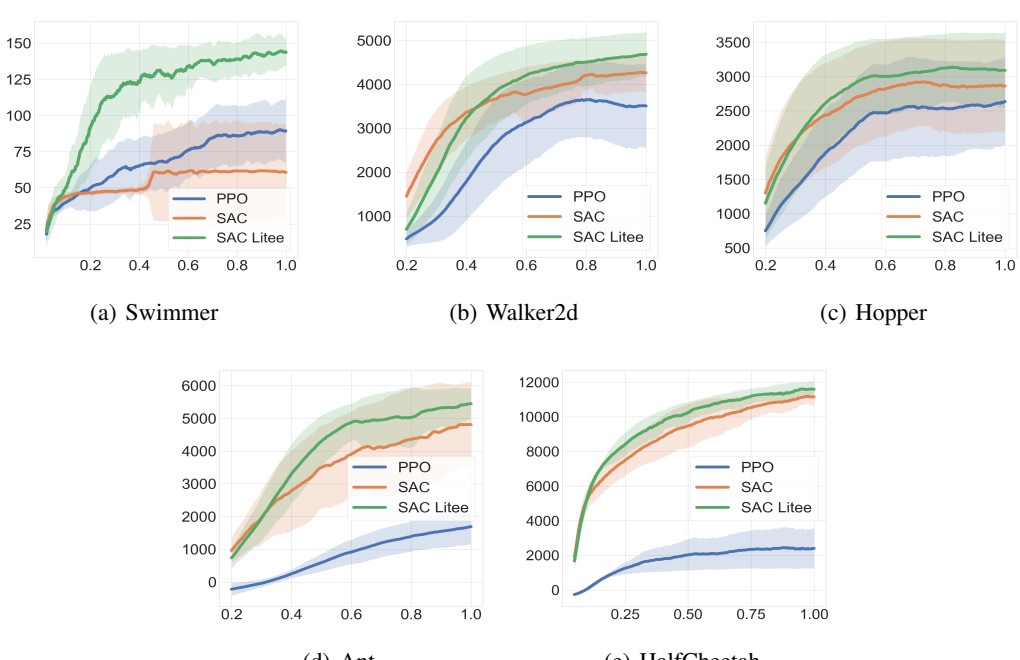

Figure 5: Experiment results on MuJoCo over seeds $1 - 5$, with the average return on the vertical axis and steps, in multiples of $1e6$, on the horizontal axis.

Since comparisons among state-of-the-art RL baselines, such as PPO, SAC, and TD3, have been extensively covered in previous studies, our focus is on investigating how *Litee* can enhance these algorithms. Thus, we concentrate on comparing the performance of each specific algorithm with and without *Litee*. In this subsection, *Litee* employs UCB-based exploration, as the Thompson Sampling-based approach has been investigated in Section 5.1.

Given that SAC achieves the best performance among existing RL algorithms on MuJoCo tasks, we investigate whether *Litee* enhances its capabilities. The results presented in Figure 5 indicate that *Litee* consistently improves the performance of SAC across various tasks. Notably, SAC combined with *Litee* demonstrates significantly better performance on the *Swimmer* task, which, although not typically considered particularly challenging, has seen limited success with SAC alone. For tasks with larger action spaces, such as *Hopper* and *Walker2d*, SAC incorporating *Litee* also achieves superior performance, as shown in Figure 5(b) and Figure 5(c).

Beyond SAC, we also investigate whether the *Litee* module can enhance the performance of other algorithms, such as PPO and TD3. The consistent performance improvements observed across multiple algorithms highlight the versatility of the *Litee* module in boosting learning efficiency and achieving better outcomes. For additional experimental results on various MuJoCo tasks involving different RL algorithms, please refer to Appendix E.

## 6 CONCLUSION

In this paper, we introduced a lightweight exploration module, *Litee*, which seamlessly integrates with existing reinforcement learning (RL) algorithms without adding extra parameters, making it computationally efficient. *Litee* utilizes the state embeddings from the RL value network to drive exploration, leaving the rest of the RL algorithm unchanged. We provided theoretical guarantees for *Litee*, establishing a sub-linear regret bound in terms of the number of interaction episodes, demonstrating its sample efficiency. For more complex tasks, we extended *Litee* to *Litee+*, incorporating a small auxiliary network to accelerate learning with only a minimal increase in parameters. Our experiments on two benchmarks, MiniHack and MuJoCo, evaluated *Litee* in both sparse and dense reward settings, and the results demonstrate that *Litee* consistently outperforms state-of-the-art baselines, bridging the gap between theoretical rigor and practical efficiency in RL exploration.

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

# A  IMPLEMENTATION

In Listing 1, we present the core code of *Litee*, while the rest of the RL algorithm remains unchanged. As shown, *Litee* is simple to implement, integrates seamlessly with any existing RL algorithm, and requires no additional parameter learning beyond what is already in the RL algorithm.

Listing 1: *Litee* core code

```
1   cov = torch.eye(256) * ridge # initialize covariance matrix
2
3   cov_inverse = torch.inverse(cov) # inverse of covariance matrix
4
5   emb = q_net.get_emb(torch.Tensor(obs), torch.Tensor(action))
6   emb = emb.squeeze().detach() # embedding of the state-action pair
7
8   bouns = torch.matmul(emb.T, torch.matmul(cov_inverse, emb))
9   bonus = np.sqrt(bonus.item()) # action-value uncertainty
10
11  reward += bonus # reshape the reward
12
13  cov += torch.outer(emb, emb) # update the covariance matrix
```

In Listing 2, we present the additional code for *Litee+* alongside that of *Litee*. As shown, *Litee+* minimizes an additional loss, specifically the inverse dynamics loss, in addition to the losses from the original RL algorithm.

Listing 2: *Litee+* additional core code

```
1   emb = q_net.get_emb(torch.Tensor(batch['obs']), torch.Tensor(batch['action'])) # embedding of state-action
        pairs in a training batch
2
3   current_emb = emb[: -1] # embeddings of the current step
4   next_emb = emb[1: ] # embeddings of the next step
5
6   predict_action = inverse_dynamic_net(current_emb, next_emb) # inferred actions
7
8   inverse_dynamics_loss = compute_inverse_dynamics_loss(predict_action, batch['action'][: -1]) # loss between
        the inferred and the executed actions
9
10  def compute_inverse_dynamics_loss(predict_action, true_action):
11      inverse_dynamics_loss=F.nll_loss(F.log_softmax(torch.flatten(predict_action, 0, 1), dim=-1), target=torch.
            flatten(true_action, 0, 1), reduction='none')
12      inverse_dynamics_loss = inverse_dynamics_loss.view_as(true_action)
13      return torch.sum(torch.mean(inverse_dynamics_loss, dim=1))
```

## B  LONG VERSION OF ALGORITHM 1

In section 3.2, algorithm 1 provides a concise version for easier comprehension. For a more thorough theoretical analysis, we present the complete version in algorithm 3. As per the standard notation in the literature on provable algorithms [19; 40], function parameters are not shared across different time steps $h \in [H]$, which is also the case in Algorithm 3. As we can see, the algorithm iteratively updates parameters $\boldsymbol{\theta}_h$ and $\boldsymbol{W}_h$, corresponding to Line 7 in algorithm 1, *i.e.*, learning the two decomposed components of the action-value function in Equation 1 by Bellman equation. Specifically, the parameter $\boldsymbol{\theta}_h$ is updated in Line 9 using its closed-form solution [22], while the extraction network $\phi_h(\cdot, \cdot)$ remains fixed. Afterwards, the extraction network $\phi_h(s, a|\boldsymbol{W}_h)$ is updated in Line 10, with the parameter $\boldsymbol{\theta}_h$ held constant. In this line, $\eta$ is the learning rate, $L_h^m$ is the Bellman loss function, and $s_h^t, a_h^t, r_h^t$ for $\forall t \in [m]$ and $\forall h \in [H]$ represent historical experiences.

---

**Algorithm 3** DQN with uncertainty

---

1: **Input:** Ridge parameter $\lambda > 0$, the exploration parameter $\alpha \geq 0$, episode length $H$, episode number $K$
2: **Initialize:** Covariance matrix $\boldsymbol{A}_h^1 = \lambda \boldsymbol{I}$, $\boldsymbol{b}_h^1 = \boldsymbol{0}$, parameters $\boldsymbol{\theta}_h^1 \sim \frac{1}{d} N(\boldsymbol{0}, \boldsymbol{I})$, networks $\phi_h^1(\cdot, \cdot|\boldsymbol{W}_h^1)$
    [39], $Q_h^1 = (\boldsymbol{\theta}_h^1)^\mathsf{T} \phi_h^1(\cdot, \cdot)$, and the target value-networks $\bar{Q}_h^1 = Q_h^1$, where $h \in [H]$

3: **for** episode $m = 1$ **to** $M$ **do**
4:     Sample the initial state of the episode $s_1^m$
5:     **for** step $h = 1, 2, ..., H$ **do**
6:         Conduct action $a_h^m = \arg\max_a Q_h^m(s_h^m, a)$ and get the next state $s_{h+1}^m$ and reward $r_h^m$

7:         Compute the target value $q_h^m = r_h^m + \max_a \bar{Q}_{h+1}^m(s_{h+1}^m, a)$

8:         Update $\boldsymbol{A}_h^{m+1} = \boldsymbol{A}_h^m + \phi_h^m(\phi_h^m)^\mathsf{T}$ and $\boldsymbol{b}_h^{m+1} = \boldsymbol{b}_h^m + q_h^m \phi_h^m$

9:         Update parameter $\boldsymbol{\theta}_h^{m+1} = (\boldsymbol{A}_h^{m+1})^{-1} \boldsymbol{b}_h^{m+1}$

10:        Update the extraction network to $\phi_h^{m+1}(\cdot, \cdot)$ with parameters $\boldsymbol{W}_h^{m+1} = \boldsymbol{W}_h^m + \eta \nabla_{\boldsymbol{w}_h^m} L_h^m$ where

$$L_h^m = \sum_{t=1}^m \sum_{h=1}^H \left| (\boldsymbol{\theta}_h^{m+1})^\mathsf{T} \phi_h^m(s_h^t, a_h^t|\boldsymbol{W}_h^m) - r_h^t - \max_a \bar{Q}_{h+1}^m(s_{h+1}^t, a) \right|^2$$

11:     Obtain UCB-based uncertainty

$$\beta_h^{m+1}(\cdot, \cdot) = \sqrt{\phi_h^{m+1}(\cdot, \cdot)^\mathsf{T} (\boldsymbol{A}_h^{m+1})^{-1} \phi_h^{m+1}(\cdot, \cdot)}$$

12:     Obtain Thompson Sampling-based uncertainty

$$\boldsymbol{\Delta\theta}_h^{m+1} \sim N(0, (A_h^{m+1})^{-1}) \Longrightarrow \beta_h^{m+1}(\cdot, \cdot) = (\boldsymbol{\Delta\theta}_h^{m+1})^\mathsf{T} \phi_h^{m+1}(\cdot, \cdot)$$

13:     Approximate the action-value function

$$Q_h^{m+1}(\cdot, \cdot) = (\boldsymbol{\theta}_h^{m+1})^\mathsf{T} \phi_h^{m+1}(\cdot, \cdot) + \alpha \beta_h^{m+1}(\cdot, \cdot)$$

14:     **end for**
15:     Update the target network $\bar{Q}_h^{m+1}(\cdot, \cdot) = Q_h^{m+1}(\cdot, \cdot), h \in [H]$
16: **end for**

---

## C    PROOF

Before delving into the detailed theory, we first review the notation used in this appendix.

Let $\pi^*$ denote the true optimal policy and $\pi_m$ represent the policy executed in episode $m \in [M]$ as outlined in Algorithm 3. The action-value and state-value functions corresponding to the policies $\pi^*$ and $\pi_m$ are represented by $Q^*$, $V^*$, and $Q^{\pi_m}$, $V^{\pi_m}$, respectively. The relationship between the state-value and action-value functions under a specific policy is given as follows:

$$V_h^*(s) = \max_a Q_h^*(s, a)$$

$$Q_h^*(s, a) = r(s, a) + \mathbb{E}_{s_{h+1} \sim \mathbb{P}_h(\cdot|s,a)} V_{h+1}^*(s_{h+1})$$

For the sake of presentation clarity, we further define several notations as follows:

$$(\mathbb{P}_h V_{h+1}^m)(s_h^m, a_h^m) = \mathbb{E}_{s_{h+1}^m \sim \mathbb{P}_h(\cdot|s_h^m, a_h^m)} V_{h+1}^m(s_{h+1}^m). \tag{11}$$

$$\delta_h^m(s_h^m, a_h^m) = r_h^m + (\mathbb{P}_h V_{h+1}^m)(s_h^m, a_h^m) - Q_h^m(s_h^m, a_h^m). \tag{12}$$

$$\zeta_h^m = V_h^m(s_h^m) - V_h^{\pi_m}(s_h^m) + Q_h^m(s_h^m, a_h^m) - Q_h^{\pi_m}(s_h^m, a_h^m). \tag{13}$$

$$\varepsilon_h^m = (\mathbb{P}_h V_{h+1}^m)(s_h^m, a_h^m) - (\mathbb{P}_h V_{h+1}^{\pi_m})(s_h^m, a_h^m) + V_{h+1}^m(s_{h+1}^m) - V_{h+1}^{\pi_m}(s_{h+1}^m). \tag{14}$$

Specifically, $\delta_h^m(s_h^m, a_h^m)$ represents the temporal-difference error for the state-action pair $(s_h^m, a_h^m)$. The notations $\zeta_h^m$ and $\varepsilon_h^m$ capture two sources of randomness, *i.e.*, the selection of action $a_h^m \sim \pi_m(\cdot|s_h^m)$ and the generation of the next state $s_{h+1}^m \sim \mathbb{P}_h(\cdot|s_h^m, a_h^m)$ from the environment.

*Proof.* **theorem 4.2**.

Based on lemma D.1, lemma D.2, and lemma D.3, we can prove theorem 4.2. Specifically, lemma D.1 decomposes the cumulative regret into three terms, where the third term is no greater than zero, then the remaining two terms are bounded by lemma D.2 and lemma D.3.

$\square$

## D    LEMMAS

**Lemma D.1.** *Adapted from Lemma 5.1 of [40]: the regret in Equation 8 can be decomposed as Equation 15, where $\langle \cdot, \cdot \rangle$ means the inner product of two vectors.*

$$\begin{aligned}
\text{Regret}_M &= \sum_{m=1}^{M} Q_1^*(s_1^m, u_1^m) - Q_1^{\pi_m}(s_1^m, a_1^m) \\
&= \sum_{m=1}^{M} V_1^*(s_1^m) - V_1^{\pi_m}(s_1^m) \\
&= \sum_{m=1}^{M} \sum_{h=1}^{H} [\mathbb{E}_{\pi^*}[\delta_h^m(s_h, a_h)|s_1 = s_1^m] - \delta_h^m(s_h^m, a_h^m)] + \sum_{m=1}^{M} \sum_{h=1}^{H} (\zeta_h^m + \varepsilon_h^m) \tag{15} \\
&\quad + \sum_{m=1}^{M} \sum_{h=1}^{H} \mathbb{E}_{\pi^*}[\langle Q_h^m(s_h, \cdot), \pi_h^*(\cdot|s_h) - \pi_m(\cdot|s_h) \rangle | s_1 = s_1^m] \\
&\leq \sum_{m=1}^{M} \sum_{h=1}^{H} [\mathbb{E}_{\pi^*}[\delta_h^m(s_h, a_h)|s_1 = s_1^m] - \delta_h^m(s_h^m, a_h^m)] + \sum_{m=1}^{M} \sum_{h=1}^{H} (\zeta_h^m + \varepsilon_h^m) \tag{16}
\end{aligned}$$

*Proof.* In Equation 15, the third equation is adapted from Lemma 5.1 of [40]. According to the definition of $\pi_m$, there is 17.

$$\langle Q_h^m(s_h, \cdot), \pi_h^*(\cdot|s_h) - \pi_m(\cdot|s_h) \rangle \leq 0 \tag{17}$$

$\square$

**Lemma D.2.** *Adapted from Lemma 5.3 of [40]: with probability at least $1 - \sigma_1$, the second term in Equation 15 can be bounded as follows:*

$$\sum_{m=1}^{M} \sum_{h=1}^{H} (\zeta_h^m + \varepsilon_h^m) \leq \sqrt{16 M H^3 \log \frac{2}{\sigma_1}} \tag{18}$$

**Lemma D.3.** *With probability at least $1 - \sigma_2$, the first term in Equation 15 can be bounded as:*

$$\sum_{m=1}^{M} \sum_{h=1}^{H} [\mathbb{E}_{\pi^*} [\delta_h^m(s_h, a_h)|s_1 = s_1^m] - \delta_h^m(s_h^m, a_h^m)] \tag{19}$$

$$\leq H \sqrt{2 M H \log \frac{2}{\sigma_2}} + C_2 \alpha H \sqrt{M d \cdot \log(1 + \frac{M}{\lambda d})}$$

$$+ \frac{C_3 \cdot H L^3 d^{\frac{5}{2}} M \sqrt{\log(\iota + \frac{1}{\sigma_2} + \frac{MA}{\sigma_2})} \|\boldsymbol{q} - \tilde{\boldsymbol{q}}\|_{\boldsymbol{H}^{-1}}}{\iota^{\frac{1}{6}}}$$

*Proof.* According to [40], there is:

$$\sum_{m=1}^{M} \sum_{h=1}^{H} [\mathbb{E}_{\pi^*} [\delta_h^m(s_h, a_h)|s_1 = s_1^m] - \delta_h^m(s_h^m, a_h^m)] \leq \sum_{m=1}^{M} \sum_{h=1}^{H} -\delta_h^m(s_h^m, a_h^m) \tag{20}$$

Considering $\delta_h^m(s_h^m, a_h^m)$, it can be decomposed as:

$$\delta_h^m(s_h^m, a_h^m) = r_h^m + (\mathbb{P}_h V_{h+1}^m)(s_h^m, a_h^m) - Q_h^m(s_h^m, a_h^m) \tag{21}$$

$$= r_h^m + (\mathbb{P}_h V_{h+1}^m)(s_h^m, a_h^m) - Q_h^*(s_h^m, a_h^m) + Q_h^*(s_h^m, a_h^m) - Q_h^m(s_h^m, a_h^m)$$

$$= \mathbb{P}_h(V_{h+1}^m - V_{h+1}^*)(s_h^m, a_h^m) + (Q_h^* - Q_h^m)(s_h^m, a_h^m)$$

$$= \underbrace{\mathbb{P}_h(V_{h+1}^m - V_{h+1}^*)(s_h^m, a_h^m) - (V_{h+1}^m - V_{h+1}^*)(s_{h+1}^m)}_{\omega_h^m}$$

$$+ \underbrace{(V_{h+1}^m - V_{h+1}^*)(s_{h+1}^m)}_{\rho_{h+1}^m} + \underbrace{(Q_h^* - Q_h^m)(s_h^m, a_h^m)}_{\varphi_h^m}$$

By Azuma-Hoeffding inequality, we can bound $\sum_{m=1}^{M} \sum_{h=1}^{H} \omega_h^m$ as Equation 22 with probability at least $1 - \sigma_3$.

$$-H \sqrt{2 M H \log \frac{2}{\sigma_3}} \leq \sum_{m=1}^{M} \sum_{h=1}^{H} \omega_h^m \leq H \sqrt{2 M H \log \frac{2}{\sigma_3}} \tag{22}$$

As $\rho_{h+1}^m$ can be decomposed as Equation 23 where $u_{h+1}^m \sim \pi_{h+1}^*(\cdot|s_{h+1}^m)$, there is Equation 24.

$$\rho_{h+1}^m = (V_{h+1}^m - V_{h+1}^*)(s_{h+1}^m) = Q_{h+1}^m(s_{h+1}^m, a_{h+1}^m) - Q_{h+1}^*(s_{h+1}^m, u_{h+1}^m) \tag{23}$$

$$\Rightarrow \sum_{m=1}^M \sum_{h=1}^H (\rho_{h+1}^m + \varphi_h^m) \tag{24}$$

$$= \sum_{m=1}^M \sum_{h=1}^{H-1} Q_{h+1}^m(s_{h+1}^m, a_{h+1}^m) - Q_{h+1}^*(s_{h+1}^m, u_{h+1}^m) + \sum_{m=1}^M \sum_{h=1}^H (Q_h^* - Q_h^m)(s_h^m, a_h^m)$$

$$= \sum_{m=1}^M \sum_{h=2}^H Q_h^*(s_h^m, a_h^m) - Q_h^*(s_h^m, u_h^m) + (Q_1^* - Q_1^m)(s_1^m, a_1^m)$$

$$\leq \underbrace{\sum_{m=1}^M \sum_{h=2}^H Q_h^*(s_h^m, a_h^m) - Q_h^*(s_h^m, u_h^m)}_{\chi} + 2H$$

Specifically, the second equation is because of $Q_{H+1}^*(s_{H+1}^m, a_{H+1}^m) = 0$ and $Q_{H+1}^m(s_{H+1}^m, a_{H+1}^m) = 0$, while the last inequality is because of $|Q_1^*| \leq H$ and $|Q_1^m| \leq H$ under the assumption that $|r(\cdot, \cdot)| \leq 1$ without loss of generality. Consequently, to complete the proof of lemma D.3, it suffices to establish a bound for $\chi$. Bounds of $\chi$ under UCB-based and Thompson Sampling-based exploration strategies are proved in Section D.1 and Section D.2, respectively. Choosing $\sigma_2 = \max\{\sigma_3, \sigma_4\}$ and $C_2 = \max\{C_2, C\}$ completes this proof. $\qquad\square$

## D.1 UCB-BASED EXPLORATION

In this subsection, we introduce the standard assumptions in the literature of *deep representation and shallow exploration* as assumption D.4, assumption D.5, and assumption D.6, which are adapted from those of [39].

**Assumption D.4.** $\|(s;a)\|_2 = 1$ *for* $\forall s \in \mathcal{S}, \forall a \in \mathcal{A}$*; and the entries of* $(s;a)$ *satisfy:*

$$(s;a)_j = (s;a)_{j+\frac{d}{2}} \tag{25}$$

**Assumption D.5.** *For* $\forall s_1, s_2 \in \mathcal{S}$ *and* $\forall a_1, a_2 \in \mathcal{A}$*, there is a constant* $l_{Lip} > 0$*, such that:*

$$\|\nabla_{\boldsymbol{W}} \phi(s_1, a_1|\boldsymbol{W}_0) - \nabla_{\boldsymbol{W}} \phi(s_2, a_2|\boldsymbol{W}_0)\|_2 \leq l_{Lip} \|(s_1;a_1) - (s_2;a_2)\|_2 \tag{26}$$

**Assumption D.6.** *The neural tangent kernel* $\boldsymbol{H}$ *of the action-value network is positive definite.*

**Lemma D.7.** *Adapted from Theorem 4.4 of [39]: suppose the standard initializations and assumptions hold. Additionally, assume without loss of generality that* $\|\boldsymbol{\theta}^*\|_2 \leq 1$*,* $\|(s_h, a_h)\|_2 \leq 1$*, and* $\|\phi(s_h, a_h)\|_2 \leq 1$*. If with the UCB-based exploration, then for any* $\sigma_4 \in (0, 1)$*, let:*

$$\alpha_h^m = \sqrt{2(d \cdot \log(1 + \frac{\iota \cdot \log A}{\lambda}) - \log \sigma_4)} + \sqrt{\lambda} \tag{27}$$

$$\eta \leq C_1 (\iota \cdot d^2 M^{\frac{11}{2}} L^6 \cdot \log \frac{MA}{\sigma_4})^{-1}; \tag{28}$$

and $\iota = \text{poly}(L, d, \frac{1}{\sigma_4}, \log \frac{MS}{\sigma_4})$ *where* $\text{poly}(\cdot)$ *means a polynomial function depending on the incorporated variables, then with probability at least* $1 - \sigma_4$, *it holds that:*

$$\chi \leq C_2 \alpha H \sqrt{Md \cdot \log(1 + \frac{M}{\lambda d})} + \frac{C_3 \cdot HL^3 d^{\frac{5}{2}} M \sqrt{\log(\iota + \frac{1}{\sigma_4} + \frac{MA}{\sigma_4})} \|\boldsymbol{q} - \tilde{\boldsymbol{q}}\|_{\boldsymbol{H}^{-1}}}{\iota^{\frac{1}{6}}} \tag{29}$$

*where* $\alpha$ *is an union bound of* $\{\alpha_1^1, ..., \alpha_H^K\}$; $C_1, C_2, C_3$ *are constants independent of the problem;* $\boldsymbol{q} = (q_1^1; q_2^1; ...; q_1^M; ...; q_H^M)$ *and* $\tilde{\boldsymbol{q}} = (Q_1^1(s_1^1, a_1^1); Q_1^1(s_2^1, a_2^1); ...; Q_1^M(s_1^M, a_1^M); ...; Q_H^M(s_H^M, a_H^M))$ *are the target and the estimated value vectors, respectively.*

Notably, the proof of the above lemma uses the concentration of self-normalized stochastic process. However, since $Q_h^m$ is not independent of $Q_h^1, Q_h^2, ..., Q_h^{m-1}$, it cannot be directly applied. Alternatively, we can adopt a similar approach to that in [40]. For simplicity of presentation, we do not explicitly handle this issue in the proof above, but it is important to keep in mind.

### D.2 THOMPSON SAMPLING-BASED EXPLORATION

**Lemma D.8.** *Under the same settings with those of lemma D.7, if with the Thompson Sampling-based exploration, Equation 30 holds, where* $C = C_2 + C_4$ *and* $C_4$ *is another problem-independent constant.*

$$\chi \leq C \alpha H \sqrt{Md \cdot \log(1 + \frac{M}{\lambda d})} + \frac{C_3 \cdot HL^3 d^{\frac{5}{2}} M \sqrt{\log(\iota + \frac{1}{\sigma_3} + \frac{MA}{\sigma_3})} \|\boldsymbol{q} - \tilde{\boldsymbol{q}}\|_{\boldsymbol{H}^{-1}}}{\iota^{\frac{1}{6}}} \tag{30}$$

*Proof.* According to Lemma A.1 of [39], $Q_h^*(s, u) - Q_h^*(s, a)$ can be decomposed as Equation 31, where $g(s, a; \boldsymbol{W}) = \nabla_{\boldsymbol{W}} \phi(s, a; \boldsymbol{W})$.

$$Q_h^*(s, u) - Q_h^*(s, a) \tag{31}$$

$$= (\boldsymbol{\theta}_h^*)^{\mathsf{T}} [\phi(s, u; \boldsymbol{W}_h^m) - \phi(s, a; \boldsymbol{W}_h^m)] + (\boldsymbol{\theta}_h^1)^{\mathsf{T}} [g(s, u; \boldsymbol{W}_h^1) - g(s, a; \boldsymbol{W}_h^1)] (\boldsymbol{W}_h^* - \boldsymbol{W}_h^m)$$

$$= (\boldsymbol{\theta}_h^1)^{\mathsf{T}} [g(s, u; \boldsymbol{W}_h^1) - g(s, a; \boldsymbol{W}_h^1)] (\boldsymbol{W}_h^* - \boldsymbol{W}_h^m)$$

$$+ \underbrace{(\boldsymbol{\theta}_h^m)^{\mathsf{T}} [\phi(s, u; \boldsymbol{W}_h^m) - \phi(s, a; \boldsymbol{W}_h^m)]}_{\vartheta_h^m} - (\boldsymbol{\theta}_h^m - \boldsymbol{\theta}_h^*)^{\mathsf{T}} [\phi(s, u; \boldsymbol{W}_h^m) - \phi(s, a; \boldsymbol{W}_h^m)]$$

According to the Thompson Sampling-based exploration in Algorithm 3, there is Equation 32.

$$(\boldsymbol{\theta}_h^m + \alpha_h^m \Delta\boldsymbol{\theta}_h^m)^{\mathsf{T}} \phi(s, u; \boldsymbol{W}_h^m) \leq (\boldsymbol{\theta}_h^m + \alpha_h^m \Delta\boldsymbol{\theta}_h^m)^{\mathsf{T}} \phi(s, a; \boldsymbol{W}_h^m) \tag{32}$$

Consequently, $\vartheta_h^m$ can be bounded as Equation 33.

$$\vartheta_h^m \leq \|\Delta\boldsymbol{\theta}_h^m\|_{\boldsymbol{A}_h^m} \|\phi(s, a; \boldsymbol{W}_h^m) - \phi(s, u; \boldsymbol{W}_h^m)\|_{(\boldsymbol{A}_h^m)^{-1}} \tag{33}$$

$$\leq (\sqrt{d} + \sqrt{2\log\frac{1}{\sigma_4}}) \|\phi(s, a; \boldsymbol{W}_h^m) - \phi(s, u; \boldsymbol{W}_h^m)\|_{(\boldsymbol{A}_h^m)^{-1}}$$

Specifically, the last inequality above is because $\Delta\boldsymbol{\theta}_h^m \sim N(0, (\boldsymbol{A}_h^m)^{-1})$. Substituting the bound of $\vartheta_h^m$ back into Equation 31 further yields:

$$Q_h^*(s, u) - Q_h^*(s, a) \tag{34}$$

$$\leq (\boldsymbol{\theta}_h^1)^\mathsf{T} \left[ g(s, u; \boldsymbol{W}_h^1) - g(s, a; \boldsymbol{W}_h^1) \right] (\boldsymbol{W}_h^* - \boldsymbol{W}_h^m)$$

$$+ (\sqrt{d} + \sqrt{2 \log \frac{1}{\sigma_4}}) \, \|\phi(s, a; \boldsymbol{W}_h^m) - \phi(s, u; \boldsymbol{W}_h^m)\|_{(\boldsymbol{A}_h^m)^{-1}}$$

$$- (\boldsymbol{\theta}_h^m - \boldsymbol{\theta}_h^*)^\mathsf{T} \left[ \phi(s, u; \boldsymbol{W}_h^m) - \phi(s, a; \boldsymbol{W}_h^m) \right]$$

Comparing Equation 34 with A.7 of [39], the difference between the regrets of Thompson Sampling-based and UCB-based exploration strategies is bounded as Equation 35, with probability at least $1 - \sigma_4$.

$$\left| \text{Regret}_{\text{Thompson Sampling}} - \text{Regret}_{\text{UCB}} \right| \leq \tag{35}$$

$$\leq \sum_{m=1}^M \sum_{h=1}^H (\sqrt{d} + \sqrt{2 \log \frac{1}{\sigma_4}}) \, \|\phi(s, a; \boldsymbol{W}_h^m) - \phi(s, u; \boldsymbol{W}_h^m)\|_{(\boldsymbol{A}_h^m)^{-1}}$$

$$+ \sum_{m=1}^M \sum_{h=1}^H \alpha_h^m \, \|\phi(s, a; \boldsymbol{W}_h^m)\|_{(\boldsymbol{A}_h^m)^{-1}} + \sum_{m=1}^M \sum_{h=1}^H \alpha_h^m \, \|\phi(s, u; \boldsymbol{W}_h^m)\|_{(\boldsymbol{A}_h^m)^{-1}}$$

$$\leq H \sqrt{Md \log(1 + \frac{M}{\lambda d})} (\sqrt{d \log(1 + \frac{M \log MA}{\lambda})} + \log \frac{1}{\sigma} + \sqrt{\lambda})$$

$$\leq C_4 \alpha H \sqrt{Md \cdot \log(1 + \frac{M}{\lambda d})}$$

Specifically, the second inequality above is based on the concentration of self-normalized stochastic processes. Similarly to the proof of UCB-based exploration, since $Q_h^m$ is not independent of $Q_h^1, Q_h^2, ..., Q_h^{m-1}$, it cannot be directly applied. However, we can alternatively adopt a similar approach to that in [40], which we do not discuss more here. $\qquad\square$

# E EXPERIMENT

## E.1 EXPERIMENT ON MINIHACK

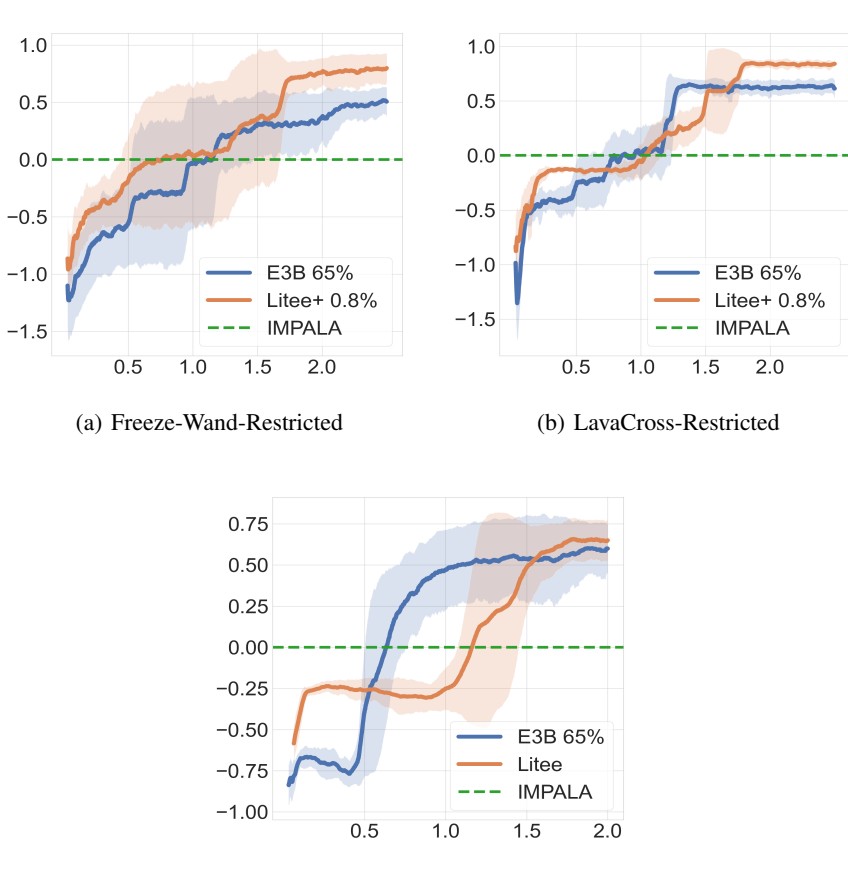

(a) Freeze-Wand-Restricted  (b) LavaCross-Restricted

(c) MultiRoom-N4-Locked

Figure 6: Additional experiment results on MiniHack.

The hyperparameters for IMPALA, E3B, and *Litee+* used in our experiments are summarized in Table 2 and Table 3, matching those used in the experiments of E3B [14]. Additionally, similar to the experiments conducted in E3B, we consider the same sixteen tasks from MiniHack; however, we present results for only a subset of these tasks. This is because E3B achieves near-perfect performance—close to the upper bound of 1 on some tasks, while others prove too difficult to complete, even for humans.

In Figure 6(a) and Figure 6(b), we present additional results for *Litee+* with Thompson Sampling-based exploration across two more tasks. The findings are consistent with those discussed in the main text. Furthermore, in Figure 6(c), we provide a comparison between E3B and *Litee*. As demonstrated, even when utilizing only *Litee* instead of *Litee+*, the performance is comparable to that of E3B, without requiring any additional parameter learning. However, it is important to note that *Litee* is capable of solving only certain tasks in MiniHack, and not all tasks can be successfully addressed by this approach.

Table 2: IMPALA Hyperparameters for MiniHack [14].

| Learning rate | 0.0001 |
|---|---|
| RMSProp smoothing constant | 0.99 |
| RMSProp momentum | 0 |
| RMSProp | $10^{-5}$ |
| Unroll Length | 80 |
| Number of buffers | 80 |
| Number of learner threads | 4 |
| Number of actor threads | 256 |
| Max gradient norm | 40 |
| Entropy Cost | 0.0005 |
| Baseline Cost | 0.5 |
| Discounting Factor | 0.99 |

Table 3: E3B and *Litee+* Hyperparameters for MiniHack.

| | Running intrinsic reward normalization | true |
|---|---|---|
| E3B and *Litee+* | Ridge regularizer | 0.1 |
| | Entropy Cost | 0.005 |
| | Exploration coefficient | 1 |
| *Litee+* | Dimension of $U$ | 256 |

### E.2 EXPERIMENT ON MUJOCO

Hyperparameters of various algorithms for the experiments on MuJoCo are completely the same with those in the public codebase CleanRL. *Litee* introduces only two more hyperparameters, *i.e.*, the exploration coefficient $\alpha$ and the ridge which is set as $\lambda = 1$. For various tasks, the exploration coefficients are summarized in Table 4. Additional experimental results on various MuJoCo tasks involving different RL algorithms can be found in Figure 7.

Table 4: Exploration coefficient for various MuJoCo tasks.

| Swimmer | 0.1 |
|---|---|
| Pusher | |
| Ant | 0.7 |
| Walker2d | 1.0 |
| Hopper | 0.4 |
| HalfCheetah | |

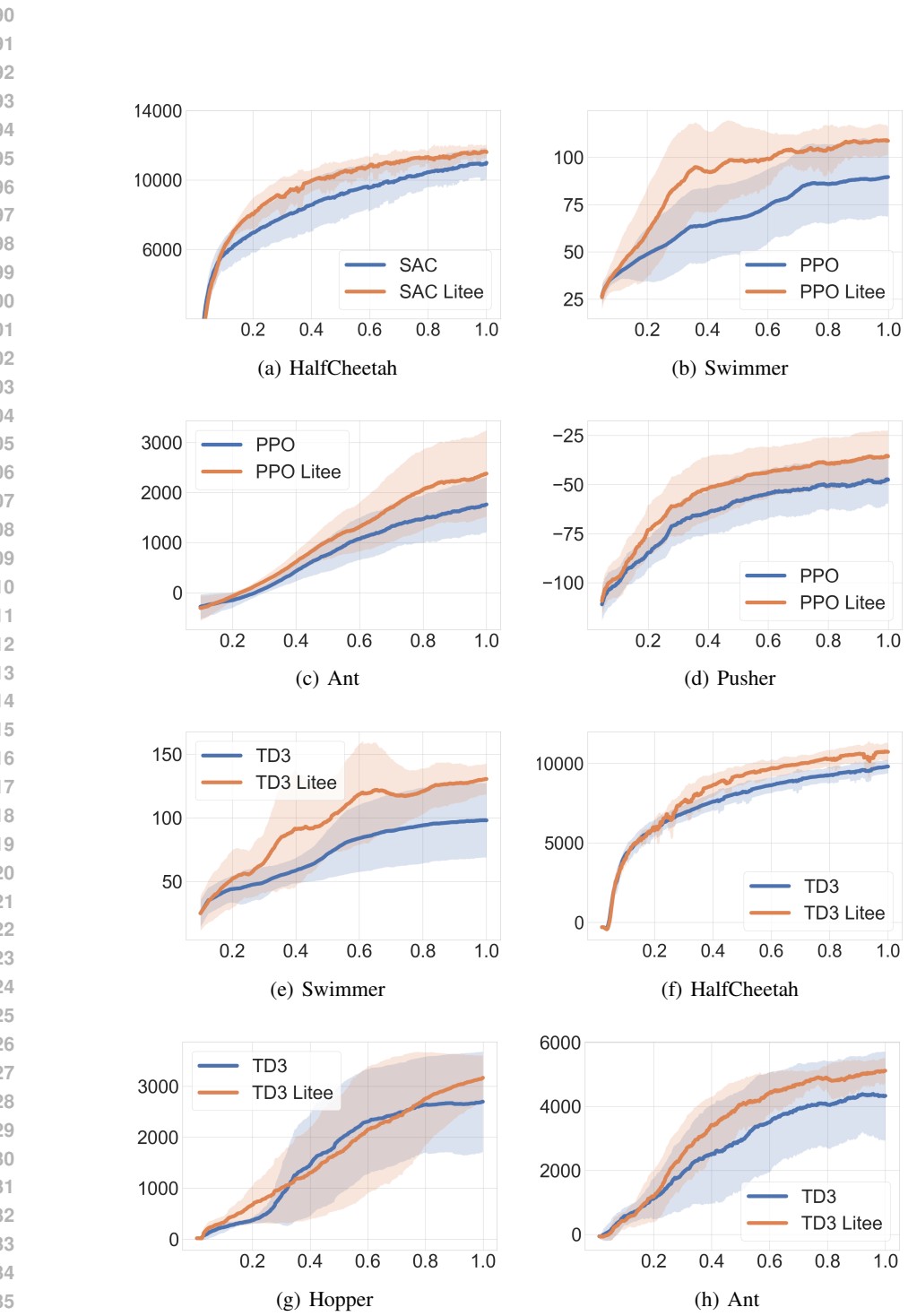

Figure 7: Additional experiment results on MuJoCo.

