# OpenReview forum: "The Critic as an Explorer: Lightweight and Provably Efficient Exploration for Deep Reinforcement Learning"
_ICLR.cc/2025/Conference — Submitted to ICLR 2025_

### Official Review · Reviewer_u6YN · 2024-10-21

**Soundness:** 3
**Presentation:** 2
**Contribution:** 3
**Rating:** 6
**Confidence:** 2

**Summary:**

This paper proposes a lightweight exploration method named Litee, which leverages linear multi-armed techniques (e.g., UCB and Thompson Sampling) as the uncertainty measurements. Theoretical analysis shows that the sample efficiency of Litee in the context of regret bound. Experimental results also show that Litee and Litee+ both achieve better sample efficiency in the dense and sparse reward settings, respectively.

**Strengths:**

1. The motivation behind the proposed method is clear and straightforward (lines 72-79).
2. This paper presents both theoretical analysis and empirical results on the sample complexity.

**Weaknesses:**

The reviewer has no concerns about the proposed method but has some suggestions on the experiments.

## Experiments
1. Although the theoretical analysis of DQN with Litee (Algorithm 1) provides the theoretical results in sample efficiency, its experimental results are needed somehow. For instance, DQN can explore nothing in Montezuma’s Revenge within limited steps. How much does Litee or Litee+ improve the DQN in this task or other tasks with sparse reward?
2. As illustrated by the authors (Section 3.3), how much does the inverse dynamics network(IDN) improve the Litee? It would be better to see the ablation studies on the sparse reward settings. For instance, the comparison of the Litee and Litee+ on MiniHack.

**Questions:**

See Weaknesses.

---

### Official Review · Reviewer_myUC · 2024-10-30

**Soundness:** 2
**Presentation:** 2
**Contribution:** 2
**Rating:** 3
**Confidence:** 5

**Summary:**

The paper proposes exploration methods for RL inspired from UCB and Thompson sampling based methods from linear bandit literature. The paper provides theoretical regret bounds for the methods as well as empirical validation.

**Strengths:**

The paper addresses a common issue in existing RL literature. The disconnection between the RL theory and deep RL literature when it comes to exploration. The paper tries to bridge this gap by proposing algorithms that are theoretically principled while being empirically scalable.

**Weaknesses:**

1. In line 72-79 and line 134-136, the paper posits that there is no RL algorithm that is both theoretically grounded as well as empirically efficient. This is not true as there have been recent works that are both empirically well performing as well as have provable theoretical guarantees (Ishfaq et al 2024a, Ishfaq et al 2024b for example).

2. The Litee algorithm builds off from Eq 2 and then utilizes UCB based bonus function and Thompson sampling. It’s not clear what’s the novelty here compared to existing RL exploration algorithms that already utilizes UCB or Thompson Sampling approach.

3. In eq 3 it uses the typical UCB bonus which are already standard in RL theory literature. For example, LSVI-UCB paper by Jin et al 2020 uses exact same bonus function. Also, $\beta(s,a)$ depends on feature parameter $\phi$. It means each time representation is updated through value network update, the bonus function needs to be recomputed based on new updated feature. Similarly, the variance matrix $A$ needs to be updated using the newly updated feature (as highlighted in Eq 5).

4. In Eq 4, for Thompson sampling based approach, it requires to take inverse of the noise variance matrix $A$, which is high dimensional matrix. Taking inverse of it is computationally non-trivial.

5. In Algorithm 2, line 10 and line 11, the Thompson sampling baed action value uncertainty estimation and reshaping reward function is essentially same as the LSVI-PHE approach described in Ishfaq et al 2021 (see Algorithm 2: LSVI-PHE for linear function approximation for example in that paper). However, the similarity with LSVI-PHE was not discussed nor the paper was cited.

6. The baselines used in the experiments are not state of the art. For example, for mujoco tasks, one of the strong baselines these days is DSAC-T from Duan et al 2023. Can you compare Litee and Litee++ with DSAC -T ?

7. Most lemmas and proof approaches are direct adaptation of existing papers for example Lemma 7. This limits the theoretical contribution of the work.

Ishfaq, Haque, et al. "Provable and Practical: Efficient Exploration in Reinforcement Learning via Langevin Monte Carlo." The Twelfth International Conference on Learning Representations. 2024a

Ishfaq, Haque, et al. "More Efficient Randomized Exploration for Reinforcement Learning via Approximate Sampling" The 1st Reinforcement Learning Conference. 2024b

Ishfaq, Haque, et al. "Randomized exploration in reinforcement learning with general value function approximation." International Conference on Machine Learning. PMLR, 2021.

Duan et al. DSAC-T: Distributional Soft Actor-Critic with Three Refinements, 2023

**Questions:**

1. Can you comment on your regret bound and how they will be if we consider linear MDP setting from Jin et al 2020?

2. How are Algorithm 1 and 2 are different from existing RL algorithms that uses UCB and Thompson sampling/randomized value function approach?

3. Can you anonymously share the full code base so that the reviewers can go through it in more detail. Just putting snippets of code in Listing 1 and Listing 2 seems insufficient to understand how Litee is implemented in the bigger picture.

---

> ### Author Response · Authors · 2024-11-25
> **Rebuttal**
>
> #### ** Q1: There have been recent works that are both empirically well performing as well as have provable theoretical guarantees.**
>
> **A1:**
>
> We thank the reviewer for suggesting additional references, and we will include them in the paper. However, these works typically assume the linearity of value functions or rely on an oracle for the embedding function $\phi$. Based on these assumptions, they propose methods, derive regret bounds, and then extend these methods directly to **deep** RL without providing further theoretical justification—despite the fact that the linearity and oracle assumptions no longer hold in such settings.
>
> The suggested study [1] also highlights the challenges of applying methods like LSVI-UCB and LSVI-PHE [2] in **deep** RL settings for practical applications. To clarify, we have included relevant statements from the literature:
>
> **Remark 6.1** [1] ... ... we use commonly used algorithms from deep RL literature as opposed to methods presented in ... ... because while these methods are provably efficient under **linear MDP settings**, in most cases, **it is not clear how to scale them to deep RL settings**. More precisely, these methods assume that a good feature is known in advance ... ... if the provided feature is not good and fixed, the empirical performance of these methods is often poor.
>
> The primary goal of our work is to address the aforementioned issue by leveraging the embedding of the value network as features. This approach enables **numerous provable methods based on linear assumptions** to be seamlessly integrated into **deep RL**, while preserving their theoretical guarantees [3]. In essence, our aim is to make rigorous, provable methods truly applicable to real-world and complex tasks. **It is not purely a theoretical paper.**
>
> **Moreover, while our focus is on UCB and Thompson Sampling, our framework is flexible enough to accommodate other exploration strategies for linear multi-armed bandits.** However, to ensure clarity and conciseness, we did not explore these additional strategies in this work and instead leave them as a direction for future research.
>
> **In summary, the objective of this work is not to develop a new exploration method with tighter theoretical bounds, but to make existing provable exploration methods truly applicable to real-world tasks.**
>
> #### **Q2: $\beta(s, a)$ depends on feature parameter $\phi$. It means each time representation is updated through value network update, the bonus function needs to be recomputed based on new updated feature.**
>
> ##### **A2:**
>
> We believe this question arises from the remark [1] we cited earlier. However, our method does not require recomputing the features or the variance matrix $A$ every time the embedding network $\phi$ is updated. As specified in Algorithm 1 and Algorithm 2, it is evident that this recomputation is unnecessary, even though the embedding network continues to be trained. This is why we consider our method lightweight, efficient, and practical.
>
> Regarding the computation of the inverse of the variance matrix $A$, as discussed in the paper, the matrix has dimensions $\tilde{d} \times \tilde{d}$, which are small. Additionally, efficient techniques such as fast rank-1 updates [4], which operate in quadratic time with respect to $\tilde{d}$, can be utilized to accelerate computation. Indeed, our experiments on MiniHack leveraged fast rank-1 updates, yielding strong performance.
>
> Moreover, simply using the embedding from the value network is inadequate for addressing sparse reward tasks, as the value network cannot be effectively trained under such conditions. This underscores the necessity of Litee$+$. Consequently, our method remains lightweight while effectively solving both sparse and dense reward tasks in **deep** RL settings.
>
> [1] Ishfaq et al. Provable and Practical: Efficient Exploration in Reinforcement Learning via Langevin Monte Carlo. ICLR 2024.
>
> [2] Ishfaq et al. More Efficient Randomized Exploration for Reinforcement Learning via Approximate Sampling. RLC. 2024.
>
> [3] Xu et al. Neural Contextual Bandits with Deep Representation and Shallow Exploration. ICLR, 2022.
>
> [4] Henaff et al. Exploration via Elliptical Episodic Bonuses. Neurips, 2022.

---

> > ### Comment · Reviewer_myUC · 2024-11-26
> >
> > I thank the authors for their response.
> >
> > As I mentioned in my original review,
> > > "In line 72-79 and line 134-136, the paper posits that there is no RL algorithm that is both theoretically grounded as well as empirically efficient. This is not true as there have been recent works that are both empirically well performing as well as have provable theoretical guarantees (Ishfaq et al 2024a, Ishfaq et al 2024b for example)."
> >
> > the manuscript claims there is no RL algorithm that are theoretically grounded as well as empirically efficient. What I am saying is this claim is not true and I would encourage the authors to include a fair discussion of these recent methods that has theoretically provable guarantee while being scalable to deep RL. The Remark the authors are quoting in their response seems to be about LSVI-UCB and their friends who are not scalable to deep RL. But my original comment was not about the algorithms that are only designed for linear MDP without any counterpart in deep RL.
> >
> > Finally most of my questions and points mentioned in the weakness section are not addressed. For example Point 1, 2, 4, 5, 6 in weaknesses section and Question 1- 3 in questions section.
> >
> > So I maintain my original score.

---

### Official Review · Reviewer_LWJ4 · 2024-10-31

**Soundness:** 2
**Presentation:** 1
**Contribution:** 2
**Rating:** 3
**Confidence:** 4

**Summary:**

This paper introduces Litee, an RL exploration algorithm designed to be lightweight. Litee utilzes the value network’s state embeddings to compute theexploration bonus without adding new parameters,hence reduces the computational complexity. Litee+, an enhanced version of Litee, integrates an auxiliary network trained with inverse dynamics to improve the exploration bonus.

**Strengths:**

1. The authors not only provide experimental results, but also provide a theoretical guarantee of the proposed algorithm, which makes the paper very comprehensive.

2. Litee and Litee+ are more parameter efficient compared to E3B

**Weaknesses:**

1. $\textbf{Lack of novelty}$: the novelty of the work is quite low, as the algorithm is almost identical to LSVI-UCB (except for the feature $\phi$ is changing over time). The theoretical framework mostly comes from [1] and [2] with some tweaks and re-organization. And Litee+ is quite related to the prior exploration RL algorithm ICM [3], and the authors do not sufficiently discuss the relations between this work and [1] [2] [3]

2. $\textbf{The experimental results are not convincing}$. The authors use part of the MuJoCo benchmark, missing HalfCheetah, Ant and Humanoid, which are most commonly used by other reinforcement learning works. SAC-Litee outperforms PPO and SAC in the swimmer task with a large margin, however, the task is too easy, which makes it a less convincing evidence that Litee is strong. Similarly, only a very small subset of tasks in MiniHack domain is used. In general, the whole set of tasks in a domain would test some aspect of an RL algorithm (exploration, credit assignment, memory, etc) comprehensively, performing well on 3 tasks is not convincing.

3. The theoretical part of the work can be better organized. In Theorem 4.2, the neural tangent kernel H comes from nowhere, and the readers will have to refer to the paper [2] in order to understand the theorem and the proof.

4. (minor)  I am not an expert of deep learning theory, but it is strange to me that the authors directly "assume" the neural tangent kernel to be $\textbf{postive-definite}$ instead of $\textbf{positive-semidefinite}$.

**Questions:**

I have mentioned most of my concerns above, in the weakness section, and I will potentially lift my rating if the authors can help me understand the following points:
1. Can authors clarify their contribution on the theoretical side of the work?
2. Can authors provide a more comprehensive benchmark results, on both MuJoCo, and MiniHack?
3. How would Litee and Litee+ perform compared to other more popular / stronger exploration algorithms on MuJoCo and MiniHack, just as used in E3B paper: E3B x RND, E3B x ICM, etc. Can the authors compare to the setting like SAC + RND, SAC + ICM?

---

> ### Author Response · Authors · 2024-11-25
> **Rebuttal**
>
> #### **Q1：The novelty of the work is quite low, as the algorithm is almost identical to LSVI-UCB.**
>
> **A1:**
>
> As outlined in the *Introduction* and *Related Works*, linear approaches rely on strong assumptions, which can significantly constrain their application in real-world tasks. Recent studies [1] further highlight the challenges of applying methods like LSVI-UCB, LSVI-PHE, etc. in deep RL settings to practical applications. For clarity, we cite relevant statements from the literature directly:
>
> **Remark 6.1** [1] ... ... we use commonly used algorithms from deep RL literature as opposed to methods presented in ... ... because while these methods are provably efficient under **linear MDP settings**, in most cases, **it is not clear how to scale them to deep RL settings**. More precisely, these methods assume that a good feature is known in advance ... ... if the provided feature is not good and fixed, the empirical performance of these methods is often poor.
>
> The primary goal of our work is to address the aforementioned issue by leveraging the embedding of the value network as features. This approach enables **numerous provable methods based on linear assumptions** to be seamlessly integrated into **deep RL**, while preserving their theoretical guarantees [2]. In essence, our aim is to make rigorous, provable methods truly applicable to real-world and complex tasks. **It is not purely a theoretical paper.**
>
> **Moreover, while our focus is on UCB and Thompson Sampling, our framework is flexible enough to accommodate other exploration strategies for linear multi-armed bandits.** However, to ensure clarity and conciseness, we did not explore these additional strategies in this work and instead leave them as a direction for future research.
>
> Furthermore, simply using the embedding from the value network is insufficient for solving sparse reward tasks, as the value network cannot be effectively trained in such scenarios. This highlights the necessity of Litee$+$. Consequently, our method remains lightweight while being capable of addressing both sparse and dense reward tasks in **deep** RL settings.
>
> #### **Q2：The authors use part of the MuJoCo benchmark, missing HalfCheetah, Ant and Humanoid.**
>
> ##### **A2:**
>
> In our paper, we reported experimental results on **five tasks** in MiniHack, with **three** presented in the main text and **two** in the appendix. Additionally, our method successfully solves all **16 tasks** in MiniHack as reported in the E3B paper [3]. We chose not to include results for simpler tasks because the baseline E3B already achieves near-perfect success rates, close to 100%, and our method performs comparably in these cases. **However, if the reviewers find it necessary, we are happy to include these results in the appendix.**
>
> We also provided additional experimental results on MuJoCo in the appendix. Following the reviewers' feedback, we have now included results for tasks such as HalfCheetah and Ant in the main text to enhance visibility, and removed those of PPO+Swimmer and TD3+Swimmer to the appendix.
>
> We did not compare our method with SAC + RND or similar approaches, as RND was specifically designed for sparse reward tasks, and we found no prior work applying it to dense reward settings. For the sparse reward tasks in MiniHack, we directly reference the results of IMPALA + RND, IMPALA + ICM, and similar methods from existing papers [1] and released codebases [1], which confirm that our method outperforms RND, ICM, and other comparable approaches.
>
> [1] Ishfaq et al. Provable and Practical: Efficient Exploration in Reinforcement Learning via Langevin Monte Carlo. ICLR 2024.
>
> [2] Xu et al. Neural Contextual Bandits with Deep Representation and Shallow Exploration. ICLR, 2022.
>
> [3] Henaff et al. Exploration via Elliptical Episodic Bonuses. Neurips, 2022.

---

> > ### Comment · Reviewer_LWJ4 · 2024-11-26
> >
> > Thanks you for your responses. However, the responses do not resolve my concerns about this work. I will keep my rating.

---

### Official Review · Reviewer_xdtx · 2024-11-04

**Soundness:** 2
**Presentation:** 2
**Contribution:** 1
**Rating:** 3
**Confidence:** 4

**Summary:**

This paper presents Litee, a lightweight algorithm for efficient exploration in reinforcement learning. Litee computes an uncertainty term from state embeddings using either UCB or Thompson Sampling techniques. To address sparse reward environments, the authors introduce Litee+, an extension that adds a minimal number of additional parameters. Compared to prior methods, Litee+ significantly reduces the parameter overhead while preserving exploration effectiveness. The paper also establishes a theoretical upper bound on the regret for both algorithms and demonstrates their effectiveness through experiments in both sparse and dense reward settings.

**Strengths:**

1. The authors present an algorithm with theoretically provable exploration efficiency and strong empirical performance. The algorithm incorporates either UCB or Thompson Sampling to calculate the uncertainty term. The exploration is achieved with only a minimal increase in parameters.
2. The authors employ an Inverse Dynamics Network (IDN) to address the sparse reward problem, achieving strong practical performance.

**Weaknesses:**

1. This approach has been extensively studied. The Litee with UCB term is nearly identical to classic algorithms for linear MDPs (e.g., https://arxiv.org/pdf/1907.05388). The novelty of Litee remains unclear.
2. The experiments are not solid. There is no table of results, and the paper only presents some plots based on three or seven repetitions, with no explanation for the variation in repetition count. Additionally, this method fails to solve all tasks in MiniHack.
3. Computational cost is not a significant constraint in RL. DRL often employs simple architectures (sometimes only a shallow MLP), and even a 100% increase in parameters to handle exploration would not be a problem for GPUs.
4. The theoretical contribution is limited. The proof has fewer than five pages and does not introduce new techniques or new messages.

**Questions:**

N/A

---

> ### Author Response · Authors · 2024-11-25
> **Rebuttal**
>
> #### **Q1: Litee with UCB term is nearly identical to classic algorithms for linear MDP.**
>
> **A1:**
>
> As outlined in the *Introduction* and *Related Works*, linear approaches rely on strong assumptions, which can significantly constrain their application in real-world tasks. Recent studies [1] further highlight the challenges of applying methods like LSVI-UCB, LSVI-PHE, etc. in deep RL settings to practical applications. For clarity, we cite relevant statements from the literature directly:
>
> **Remark 6.1** [1] ... ... we use commonly used algorithms from deep RL literature as opposed to methods presented in ... ... because while these methods are provably efficient under **linear MDP settings**, in most cases, **it is not clear how to scale them to deep RL settings**. More precisely, these methods assume that a good feature is known in advance ... ...
>
> The primary goal of our work is to address the aforementioned issue by leveraging the embedding of the value network as features. This approach enables **numerous provable methods based on linear assumptions** to be seamlessly integrated into **deep RL**, while preserving their theoretical guarantees [2]. In essence, our aim is to make rigorous, provable methods truly applicable to real-world and complex tasks.
>
> **Moreover, while our focus is on UCB and Thompson Sampling, our framework is flexible enough to accommodate other exploration strategies for linear multi-armed bandits.** However, to ensure clarity and conciseness, we did not explore these additional strategies in this work and instead leave them as a direction for future research.
>
> Furthermore, simply using the embedding from the value network is insufficient for solving sparse reward tasks, as the value network cannot be effectively trained in such scenarios. This highlights the necessity of Litee$+$. Consequently, our method remains lightweight while being capable of addressing both sparse and dense reward tasks in **deep** RL settings.
>
> #### **Q2: The paper only presents some plots based on three or seven repetitions, with no explanation for the variation in repetition count. Additionally, this method fails to solve all tasks in MiniHack.**
>
> ##### **A2**:
>
> In DRL research, it is standard practice to compare model performance using the mean and standard deviation across 3–7 seeds [3, 4, 5, 6 ... ... and **many**] . We are unclear about the comment regarding **"no explanation for the variation in repetition count"**, as this setting is widely accepted in the field.
>
> Additionally, our method successfully solves all **16 tasks** in MiniHack as reported in the E3B paper [3]. We chose not to report results for simpler tasks because the baseline method already achieves near-perfect success rates, close to 100%, and our method performs comparably in these cases. To maintain focus, we reported only the tasks where our approach demonstrates significant improvements. **If the reviewer feels it is necessary, we are willing to include the omitted results in the appendix.**
>
> [1] Ishfaq et al. Provable and Practical: Efficient Exploration in Reinforcement Learning via Langevin Monte Carlo. ICLR 2024.
>
> [2] Xu et al. Neural Contextual Bandits with Deep Representation and Shallow Exploration. ICLR, 2022.
>
> [3] Haarnoja et al. Soft Actor-Critic: Off-Policy Maximum Entropy Deep Reinforcement Learning with a Stochastic Actor. ICML, 2018.
>
> [4] Schulman et al. Proximal Policy Optimization Algorithms. arXiv, 2017.
>
> [5] Henaff et al. Exploration via Elliptical Episodic Bonuses. Neurips, 2022.
>
> [6] Espeholt et al. IMPALA: Scalable Distributed Deep-RL with Importance Weighted Actor-Learner Architectures. ICML, 2018.

---

> ### Author Response · Authors · 2024-11-25
> **Rebuttal**
>
> #### **Q3: Computational cost is not a significant constraint in RL. DRL often employs simple architectures.**
>
> **A3:**
>
> Computational cost is a critical factor in practical tasks, as it directly impacts both efficiency and scalability. In response to the assertion that **"DRL often employs simple architectures"**, this claim does not hold true for practical applications. When states include multi-modal data like images and texts, etc. small and simple network architectures are insufficient. Instead, processing such states effectively demands substantial computational resources, as demonstrated in Table 1 of our paper.
>
> Numerous successful DRL implementations, such as AlphaZero and Atari models, employ networks with tens of layers and tens to hundreds of millions of parameters. Moreover, several studies [7] have demonstrated that more complex network architectures can improve the robustness and performance of DRL models. As emphasized earlier, **since this is not purely a theoretical paper, our primary focus is on making the method efficient and applicable to real-world, complex tasks**.
>
> #### **Q4. The theoretical contribution is limited. The proof has fewer than five pages and does not introduce new techniques or new messages.**
>
> **A4：**
>
> We acknowledge the brevity of the proof; however, we believe that the five pages allocated are sufficient to cover the theoretical analysis presented in our paper. **It is important to emphasize that this is NOT a purely theoretical RL paper.** Rather, our work aims to bridge the gap between theoretical rigor and practical efficiency in **deep** RL exploration. The theoretical analysis ensures that, even as we relax the linearity assumption of value functions in RL to make the method practically applicable, the desirable theoretical property—sublinear regret—remains intact.
>
> **As noted above, while our primary focus is on UCB and Thompson Sampling, our framework is versatile enough to support other exploration strategies for linear multi-armed bandits.** This flexibility allows for the development and proof of new exploration methods within this framework; however, proposing such methods is beyond the scope of this work.
>
> [7] Schwarzer et al. Bigger, Better, Faster: Human-Level Atari with Human-Level Efficiency. ICML, 2023.

---

> > ### Comment · Reviewer_xdtx · 2024-12-03
> >
> > Thank you for your response. However, it did not address my concerns. I will keep my current score.

---

### Meta-Review · Area_Chair_nUyT · 2024-12-21

**Metareview:**

This paper studies lightweight and provably efficient exploration techniques for deep reinforcement learning, introducing the Litee and Litee+ algorithms with theoretical regret bounds and empirical results. The paper achieves reduced computational complexity and parameter overhead using UCB and Thompson Sampling-based techniques. However, the reviewers found the novelty of the work limited, with substantial overlap with existing methods like LSVI-UCB and ICM, and noted that the experimental results lacked robustness and comprehensive benchmarking. During the rebuttal phase, the authors' responses did not sufficiently address these concerns, leaving the reviewers unconvinced about the contribution and practical significance of the work. Overall, the recommendation for this paper is rejection.

**Additional Comments On Reviewer Discussion:**

During the rebuttal period, the reviewers raised several concerns regarding the novelty, experimental robustness, and theoretical contributions of the paper. Reviewer xdtx highlighted that the proposed algorithms closely resemble existing methods like LSVI-UCB and questioned the experimental design, including inconsistent repetition counts and missing results for key benchmarks. Reviewer LWJ4 emphasized the lack of comparisons to stronger baselines, limited task coverage in MiniHack and MuJoCo, and theoretical contributions largely adapted from prior work. Reviewer myUC noted insufficient novelty, computational challenges in the proposed methods, and an inadequate discussion of related works with theoretical guarantees. Reviewer u6YN requested additional ablation studies and evidence on sparse reward tasks. The authors addressed these by clarifying distinctions from related methods, updating experimental results to include additional tasks, and committing to include omitted baselines and references. However, the responses did not convincingly resolve concerns about novelty, experimental rigor, or the limited scope of the theoretical framework. Balancing these unresolved issues with the strengths of the proposed techniques, the final decision was to recommend rejection.

---

### Decision · Program_Chairs · 2025-01-22

Reject